# Comparative Transcriptomic Analysis of mRNAs, miRNAs and lncRNAs in the *Longissimus dorsi* Muscles between Fat-Type and Lean-Type Pigs

**DOI:** 10.3390/biom12091294

**Published:** 2022-09-13

**Authors:** Jian Zhang, Jiying Wang, Cai Ma, Wenlei Wang, Heng Wang, Yunliang Jiang

**Affiliations:** 1Shandong Provincial Key Laboratory of Animal Biotechnology and Disease Control and Prevention, Shandong Agricultural University, No. 61 Daizong Street, Taian 271018, China; 2Shandong Key Laboratory of Animal Disease Control and Breeding, Institute of Animal Science and Veterinary Medicine, Shandong Academy of Agricultural Sciences, Jinan 250100, China

**Keywords:** pig, *Longissimus dorsi* muscle, transcriptome, mRNA, miRNA, lncRNA, intramuscular fat

## Abstract

In pigs, meat quality and production are two important traits affecting the pig industry and human health. Compared to lean-type pigs, fat-type pigs contain higher intramuscular fat (IMF) contents, better taste and nutritional value. To uncover genetic factors controlling differences related to IMF in pig muscle, we performed RNA-seq analysis on the transcriptomes of the *Longissimus dorsi* (LD) muscle of Laiwu pigs (LW, fat-type pigs) and commercial Duroc × Landrace × Yorkshire pigs (DLY, lean-type pigs) at 150 d to compare the expression profiles of mRNA, miRNA and lncRNA. A total of 225 mRNAs, 12 miRNAs and 57 lncRNAs were found to be differentially expressed at the criteria of |log_2_(foldchange)| > 1 and *q* < 0.05. The mRNA expression of *LDHB* was significantly higher in the LD muscle of LW compared to DLY pigs with log_2_(foldchange) being 9.66. Using protein interaction prediction method, we identified more interactions of estrogen-related receptor alpha (*ESRRA*) associated with upregulated mRNAs, whereas versican (*VCAN*) and proenkephalin (*PENK*) were associated with downregulated mRNAs in LW pigs. Integrated analysis on differentially expressed (DE) mRNAs and miRNAs in the LD muscle between LW and DLY pigs revealed two network modules: between five upregulated mRNA genes (*GALNT15*, *FKBP5*, *PPARGC1A*, *LOC110258214* and *LOC110258215*) and six downregulated miRNA genes (*ssc-let-7a*, *ssc-miR190-3p*, *ssc-miR356-5p*, *ssc-miR573-5p*, *ssc-miR204-5p* and *ssc-miR-10383*), and between three downregulated DE mRNA genes (*IFRD1*, *LOC110258600* and *LOC102158401*) and six upregulated DE miRNA genes (*ssc-miR1379-3p*, *ssc-miR1379-5p*, *ssc-miR397-5p*, *ssc-miR1358-5p*, *ssc-miR299-5p* and *ssc-miR1156-5p*) in LW pigs. Based on the mRNA and ncRNA binding site targeting database, we constructed a regulatory network with miRNA as the center and mRNA and lncRNA as the target genes, including *GALNT15/ssc-let-7a/LOC100523888, IFRD1/ssc-miR1379-5p/CD99*, etc., forming a ceRNA network in the LD muscles that are differentially expressed between LW and DLY pigs. Collectively, these data may provide resources for further investigation of molecular mechanisms underlying differences in meat traits between lean- and fat-type pigs.

## 1. Introduction

Pork is a vital source of protein, energy and iron for humans, and occupies an important position in the world meat consumption [1]. Meat quality is attracting more and more attention in recent years. Intramuscular fat (IMF) content is a primary pork quality indicator, affecting meat quality traits including tenderness, juiciness, flavor and nutritional value [2]. IMF refers to the fat extracted from muscle tissue, which is deposited in muscle fiber, outer membrane and inner membrane and is considered to be a late developing fat storage during the process of fat deposition [3]. Meat quality is also influenced by the composition of different types of skeletal muscle fibers including type I (slow/oxidative), type IIa (fast/oxido-glycolic), IIX (between type IIa and IIb) and type IIb (fast/glycolic) [4]. Type I muscle fibers can carry out long-lasting slow activity, as they have more capillaries, lipids, myoglobin than type II muscle, while type II muscle fibers are more capable of rapid muscle contraction because they contain more glycolytic enzymes [5]. The relatively lower expression of *MyHC-IIb* gene and lower proportion of type IIb fiber is related to the better meat quality of Laiwu (LW) and Jinhua pigs [6].

Compared with traditional lean-type pig breeds, the fat-type breeds contain higher IMF contents, better taste and nutritional value [7,8]. Nevertheless, lean-type pigs have higher lean rates and growth rates than fat-type pigs [9]. Several studies reported the comparative analysis in the transcriptome between lean-type and fat-type pigs, such as Large White vs. Basque pigs [10], Wei vs. Yorkshire pigs [11], Polish Landrace vs. Puławska pigs [12] as well as the proteome of Lantang and Landrace pigs [13] and the transcriptome and proteome of Pietrain and Duroc pigs [14]. Laiwu pigs (LW) are a classical fat-type indigenous pig breed distributed in Shandong province of China, well known for its extremely higher IMF content (about 10%) [15] than commercial Duroc × Landrace × Yorkshire (DLY) pigs, which are lean-type pigs showing higher lean meat rate and faster growth rate [16]. Two studies analyzed the molecular mechanisms underlying IMF deposition in LW pigs by transcriptome, DNA methylome [17] and proteome approaches [18]; however, the causative gene(s) remains to be determined.

With the rapid development of high-throughput sequencing, numerous non-coding RNAs (ncRNAs) including microRNA (miRNA) and long noncoding RNA (lncRNA) have been found to play important roles in IMF deposition and muscle development [19]. MicroRNAs are a family of post-transcriptional gene repressors, which are widely involved in the regulation of gene expression in various biological processes associated with fat synthesis, deposition and metabolism [20]. Competing endogenous RNA (ceRNA) hypothesis postulated that various types of mRNAs, lncRNAs, circRNAs (circular RNAs) and pseudogenes could compete for the same microRNA response elements to mutually regulate gene expression [21]. In one previous study, the potential lncRNAs/circRNAs-miRNAs-mRNAs regulatory networks shared *MYOD1*, *PPARD*, *miR-423-5p* and *miR-874*, which were associated with skeletal muscle muscular proliferation, differentiation/regeneration and adipogenesis [22].

In this study, the transcriptomes of the *Longissimus dorsi* muscle were analyzed to identify differences in gene expression profiles of mRNA, miRNA and lncRNA between LW pigs and DLY pigs, to uncover miRNA-mRNA interaction pathways and to construct mRNA-miRNA-lncRNA interaction networks. The data provided in this study may assist in deciphering the molecular mechanism underlying IMF deposition and other meat quality traits in pigs.

## 2. Materials and Methods

### 2.1. Animals and Tissue Samples

A total of three castrated male Laiwu pigs with unrelated genetic background reared under identical feeding conditions and environment were randomly selected from the Conservation Center of Laiwu pigs (Laiwu, Shandong, China) at 150 d. Similarly, three castrated male DLY pigs of 150 d of age were randomly selected from Beihe Pig Breeding Co., Ltd. (Laiwu, Shandong, China). The animals were allowed access to food and water *ad libitum* and were housed under identical conditions. To minimize animal suffering, the pigs were weighed and electrically stunned before death. Immediately after slaughter, about 30 g sample of the *Longissimus dorsi* muscle (LD) at the third lumbar vertebra of each pig was collected into a 2 mL cryogenic vial (Corning, NY, USA) and frozen in liquid nitrogen for further study [15,23]. For RNA-seq, samples were transported in dry ice to BGI (Shenzhen, China) and stored at −80 °C for total RNA extraction. The same samples were also used for real-time quantitative PCR (RT-qPCR). All animal care and experimental procedures were in accordance with the guidelines for the care and use of laboratory animals prescribed by Shandong Agricultural University and the Ministry of Agriculture of China (No. SDAUA-2021-097).

### 2.2. IMF Content Measurement

The IMF content in the LD muscle tissues of pigs was measured by Soxhlet petroleum ether extraction method [24]. Briefly, after removing the fascia, blood vessels and connective tissue, skeletal muscle sample was ground using a meat grinder. Ten grams of the sample was dried to a constant weight in a drying oven (Yiheng Scientific Instrument Co., Ltd., Shanghai, China) at 103 °C. Dried meat sample with filter paper package was treated with an appropriate amount of petroleum ether into the Soxhlet bottle and extracted continuously for 6–8 h. Then, the sample was placed in a desiccator overnight and placed in a vacuum drying oven at 80 °C for 12 h and finally weighed. The IMF was calculated by dividing (the sample of dried meat wrapped in the filter paper (g) minus the weight of the dried meat of the filter paper after extraction (g)) by the sample of dried meat (g) multiplied by 100.

### 2.3. Total RNA Isolation, Library Preparation, and Sequencing

Total RNA was extracted from the LD muscle tissues using Trizol (Invitrogen, Carlsbad, CA, USA) according to manufacturer’s instructions. The quantity and purity of the total RNA were evaluated using Agilent 2100 Bioanalyzer (Agilent Technologies, Santa Clara, CA, USA). Total RNA was divided into two samples after preparation, and each was used for mRNA and lncRNA cDNA library and miRNA library construction, respectively. For mRNA and lncRNA cDNA library, the first step involves the removal of ribosomal RNA (rRNA) using target-specific oligos and RNase H reagents. The cleaved RNA fragments were copied into first strand cDNA using reverse transcriptase (MGIEasy RNA, Shenzhen, China) and random primers, followed by second strand cDNA synthesis using DNA Polymerase I (MGIEasy RNA, Shenzhen, China) and RNase H (MGIEasy RNA, Shenzhen, China). The products were enriched with PCR to create the final cDNA library. For miRNA cDNA library, total RNA was purified to obtain 18–30 nt small RNA, which was ligated to a 5′-adaptor and a 3′-adaptor. The adaptor-ligated small RNAs were subsequently transcribed into cDNA by SuperScript II Reverse Transcriptase (Invitrogen, Carlsbad, CA, USA) and then PCR amplified to enrich the cDNA fragments. The library quality was checked by analyzing the distribution of the fragments size using the Agilent 2100 bioanalyzer, and quantified by RT-qPCR. The qualified libraries were sequenced pair end on the BGISEQ-500 platform (BGI, Shenzhen, China).

### 2.4. Bioinformatic Analysis of DE mRNA, miRNA and lncRNA and Their Interactions

The raw sequencing data from mRNA, miRNA and lncRNA libraries were filtered with SOAPnuke 1.5.2 (https://github.com/BGI-flexlab/SOAPnuke, accessed on 1 May 2021) [25]. In this step, reads containing sequencing adaptor, low-quality reads (base quality less than or equal to 5) and unknown base (‘N’ base) were removed. After filtering, clean reads were obtained and stored in FASTQ format. The clean reads were mapped to the *Sus scrofa* reference genome (Sscrofa11.1, https://www.ncbi.nlm.nih.gov/assembly/GCF_000003025.6, accessed on 1 May 2021) using HISAT2 2.0.4 (http://www.ccb.jhu.edu/software/hisat/index.shtml, accessed on 1 May 2021) [26]. The clean tags of miRNA were mapped to the reference genome with Bowtie2 (http://bowtie-bio.sourceforge.net/bowtie2/index.shtml, accessed on 1 May 2021) [27]. FPKM (reads per kilo base of the exon model per million mapped reads) values obtained using Cufflinks 2.1.1 (http://cole-trapnell-lab.github.io/cufflinks/, accessed on 1 May 2021) were used as values for normalized gene expression [28] and annotated with NCBI genome assembly [29].

The statistically significant differentially expressed (DE) mRNAs, miRNAs and lncRNAs were obtained by a *q* value threshold of <0.05 and |log_2_ (fold change)| >1 using the DEseq2 software (v1.36, Simon Anders, Heidelberg, Germany) [30]. Analyses of the differentially expressed mRNAs by GO were performed using Gene Ontology database (http://www.geneontology.org/, accessed on 1 May 2021) [31,32], and KEGG pathway analysis was performed by KEGG database [33]. Pathway enrichment statistical analyses were performed using R package phyper (https://stat.ethz.ch/R-manual/R-devel/library/stats/html/Hypergeometric.html, accessed on 1 May 2021) to calculate *p* value [34]. Multiple testing correction of *p* value was applied using R package qvalue (https://bioconductor.org/packages/release/bioc/html/qvalue.html, accessed on 1 May 2021) with corrected *q* values < 0.05 by Bonferroni [35]. The plots of KEGG enrichment were prepared using R package ggplot2 (https://www.rdocumentation.org/packages/ggplot2/versions/3.3.6, accessed on 1 May 2021) [36]. The heatmap was drawn by pheatmap R package (https://cran.r-project.org/web/packages/pheatmap/index.html, accessed on 1 May 2021) according to the gene expression level in different samples. A protein–protein interaction network was constructed by the Search Tool for the Retrieval of Interacting Genes (STRING) online software against the *Sus scrofa* database. Additionally, Qiagen’s Ingenuity Pathway Analysis (IPA) was used to identify the predicted upstream regulators of DEGs.

We constructed ncRNAs regulatory network to reveal the interaction between ncRNAs and mRNAs in porcine LD muscles. The interaction of miRNAs with mRNAs and lncRNAs was predicted with miRDB (http://www.mirdb.org/, accessed on 1 May 2021), miRcode (http://www.mircode.org/index.php, accessed on 1 May 2021) and TargetScan (http://www.targetscan.org/vert_71/, accessed on 1 May 2021).The prediction of target lncRNAs of mRNAs was performed using RNAplex [37]. The ncRNA networks and lncRNA or miRNA interactions of interest were visualized by Cytoscape (V.3.8.2).

### 2.5. Immunohistochemistry

The LD muscle tissue sections from LW and DLY pigs were processed according to standard protocols following the procedures as described by Zhang et al. [38]. Mouse monoclonal antibody to S46 (myosin heavy chain, slow developmental; 1:200, AB_528376, DSHB, Iowa city, IA, USA) and mouse monoclonal antibody to F59 (myosin heavy chain, all fast isoforms; 1:200, AB_528373, DSHB, Iowa city, IA, USA) were used to incubate the above prepared sections. After overnight storage at 4 °C, the sections were incubated with goat anti-mouse IgG (PV-9002, ZSGB-Bio Co., Ltd., Beijing, China) for 1 h in a 37 °C thermostat box. The sections were washed with PBS (3 × 5 min) after these incubations. All samples were incubated in diaminobenzidine tetrachloride (DAB kit, PN3122, G-CLONE Co., Ltd., Beijing, China) for 1~3 min and counterstained with hematoxylin and color developed in tap water. The sections were then dehydrated, sealed in clear resin, mounted, and observed microscopically for the distribution of positive cells using a bright field of view.

### 2.6. Overexpression and Knockdown Assay

The entire coding region of murine *GALNT15* gene was amplified, and polymerase Gflex (TaKaRa, Dalian, China) was used to ensure high fidelity (Appendix A). The polymerase chain reaction (PCR) fragments were generated by double enzyme (*Hind* III and *Xho* I) digestion and ligated with pcDNA3.1(+) expression vectors (Invitrogen, Carlsbad, CA, USA) by T4 DNA ligase (Thermo Fisher Scientific, Waltham, MA, USA), which were transformed into DH5α (TaKaRa, Dalian, China) competent cells. After being confirmed by bidirectional sequencing and purified using an EndoFree Plasmid Midi Kit (Aidlab, Beijing, China), these plasmids were used for transfecting 3T3-L1 cells. The obtained plasmid was named pcDNA3.1(+)-GALNT15. Empty pcDNA3.1(+) vector was used as the control.

For knockdown (KD) assay, siRNA was designed according to murine *GALNT15* mRNA (Shanghai GenePharma Co., Ltd., Shanghai, China). The negative control of the siRNA had the same composition with the siRNA sequence but had no homology with *GALNT15* mRNA (Appendix A). Three pairs of siRNAs were designed. The most effective siRNA was used to analyze the knockdown effect of siRNA on *GALNT15* gene.

### 2.7. Cell Culture and Transfection

The 3T3-L1 cells were planted in a 24-well culture growth (pyruvate-free) DMEM (Gibco, Camarillo, CA, USA) medium supplemented with 10% fetal bovine serum (Biological Industries, Kibbutz Beit-Haemek, Israel) and penicillin–streptomycin (100 U/mL penicillin, 100 mg/L streptomycin) at 37 °C in a humidified atmosphere (5% CO_2_, 95% air). 3T3-L1 cells were transfected with pcDNA3.1(+)-GALNT5 overexpression plasmid or siRNA when grown to 70~80% confluency using Lipofectamine 3000 (Invitrogen, Carlsbad, CA, USA) according to the manufacturer’s instruction. RT-qPCR and Oil Red O (ORO) staining were performed on the transfected cells. Lipid droplet generation of 3T3-L1 cells and sections were visualized using Oil Red O (Beyotime, Shanghai, China) staining.

### 2.8. RT-qPCR

The total RNA used for above transcriptome analysis was also used for RT-qPCR. The cDNA was synthesized using a Primescript RT Mix Kit with gDNA Clean (Accurate Biotechnology, Changsha, China), and the resultant cDNA was stored at −20 °C for mRNA and lncRNA expression analysis. Synthesis of the cDNA of miRNA was performed using a miRNA 1st strand cDNA synthesis kit (Accurate Biotechnology, Changsha, China) according to the manufacturer’s protocol. Specific primers of DE mRNAs and DE ncRNAs were designed based on the gene sequences of pigs reported in GenBank using DNAMAN 8.0 and synthesized by Shanghai Bioengineering Co., Ltd., Shanghai, China.

The qPCR for mRNAs and lncRNAs was conducted by mixing 10 μL of 2×SYBR Green Pro Taq HS Premix (Accurate Biotechnology, Changsha, China), 0.4 μL of each primer (forward and reverse, Appendix A), 100 ng of cDNA template and RNase free water up to 20 μL and run on a Light Cycler 96 real-time PCR system (Roche, Basel, Switzerland) with the following program: 95 °C for 30 s, followed by 40 cycles of denaturation at 95 °C for 5 s and annealing and extension at gene-specific temperature (Appendix A) for 30 s. The qPCR for miRNA was conducted by mixing 10 μL of 2X SYBR Green Pro Taq HS Premix II (Accurate Biotechnology, Hunan, China), 0.4 μL of miRNA specific primer (Appendix A), 0.4 μL miRNA qPCR 3′ primer (Accurate Biotechnology, Changsha, China), 100 ng of cDNA template and RNase free water up to 20 μL and run on a Light Cycler 96 real-time PCR system (Roche, Basel, Switzerland) with the following program: 95 °C for 30 s, followed by 40 cycles of denaturation at 95 °C for 5 s and annealing and extension at annealing temperature (Appendix A) for 30 s. The melting curves were obtained, and quantitative analysis of the data was performed using the 2^−ΔΔ*CT*^ relative quantification method [39]. The *GAPDH* and *U6* were used as internal reference gene for the relative quantification of mRNA and lncRNA, and miRNA, respectively.

### 2.9. Statistical Analysis

Statistical analysis was performed using Mann–Whitney U test to compare the live weight and IMF content of LW and DLY pigs (*n* = 3). Statistical analysis was performed using Student’s *t* test to compare the mRNA expression level of the genes, each experiment was repeated four times, and at least three independent experiments were performed. The quantitative data were presented as the mean ± SEM and the *p* value below 0.05 is considered as significant. Correlation analysis and plot between RT-qPCR and RNA-seq was performed using GraphPad Prism 8.0. *p* value below 0.05 is considered as significant.

## 3. Results

### 3.1. Overview of the Transcriptome Sequencing Data of Porcine LD Muscles

At 150 d, the live weight of DLY pigs was significantly higher than that of LW pigs (*p* < 0.05, Figure 1A), while the IMF content in the LD muscle of DLY pigs was significantly lower than that of LW pigs (*p* < 0.05, Figure 1B). Oil red O staining showed that the *Longissimus dorsi* muscle of LW pigs contained higher fat deposition than that of DLY pigs (Figure 1C) and immunohistochemical analysis showed that there were a large number of F59 (fast/type2/glycolytic) myofibers and less F46 (slow/type1/oxidative) myofibers in the muscle tissue of DLY pigs compared to LW pigs (Figure 1D). Three LD muscle tissue samples collected from each of these LW and DLY pigs were used for transcriptome sequencing to identify genes underlying IMF variation and meat quality traits. For each sample, an average of 114.66 million paired-end clean reads and 11.46 Gb of clean data were produced for mRNA and lncRNA cDNA libraries on DNBseq platform. All six samples had at least 90.68% reads equal to or exceeding Q30 and the clean reads ratio of each sample was greater than 92.93% after removing adapters, low quality reads and data containing N, and the total mapping ratio of each sample was greater than 92.38% (Table 1). For miRNA sequencing, an average of 23.89 million clean tag counts of miRNA cDNA libraries were obtained for each LD sample, with more than 98.5% equal to or exceeding Q20, and the total mapping ratio of each sample was greater than 89.07% (Table 2). Furthermore, a total of 30,681 expressed genes were mapped to pig genome, including 20,727 mRNAs, 6702 lncRNAs and 3252 miRNAs. These sequence data were uploaded to NCBI with the accession number PRJNA815878.

### 3.2. Differentially Expressed mRNAs in the LD Muscle between LW and DLY Pigs

Compared with DLY pigs, a total of 225 differentially expressed (DE) mRNAs (105 upregulated and 120 downregulated) were revealed in the LD muscle of LW pigs (Figure 2A). According to the *q* value threshold of <0.05 and log_2_ (foldchange) > 3, the upregulated and downregulated DE mRNAs in the LD muscle of LW pigs were shown in Table 3 and Table 4, respectively. The greatest fold change in mRNA expression was *LDHB* (upregulated) (Table 3) and *LOC110258854* (downregulated) (Table 4), respectively. According to the expression level, all of the DE mRNAs in porcine LD muscle were clustered in two clades of LW and DLY as visualized by the heatmap plot (Figure 2B). The upregulated and downregulated DE mRNAs between LW and DLY pigs were further analyzed by GO and KEGG pathway approaches. For the upregulated DE mRNAs expressed in the LD muscle of LW pigs, GO terms were mainly enriched in chromatin DNA binding, nucleoside binding and aromatase activity, organelle membrane, glycerol biosynthetic process, inosine catabolic process and fatty acid homeostasis (Figure 2C), and KEGG pathways were related to processes required for metabolic pathways, glucagon signaling and adipocytokine signaling pathways (Figure 2D). For the downregulated DE mRNAs expressed in the LD muscle of LW pigs, GO terms were mainly enriched in actin-binding and DNA integration (Figure 2E), and KEGG pathways were related to processes required for metabolic pathways and MAPK signaling pathway (Figure 2F). Ten of the randomly selected DE mRNAs were validated by RT-qPCR, showing a significant correlation of 0.857 (*p* < 0.01) (Figure 2G). Metabolic pathway plays a critical role in skeletal muscle contraction, development and IMF deposition [40]. The DE mRNAs enriched into metabolic pathways include *PTGES2*, *PNP*, *GALNT15*, *KYAT1*, *NNT*, *LDHB*, *ST8SIA5*, *GOT1*, *PLA2G4E*, *PLPP1*, *TKTL2*, *DHCR24*, *MTHFD1*, *AK5*, *LOC106505238*, etc. (Figure 2H).

A network of protein–protein interactions (PPI) was constructed to reveal the interactions among DE mRNA genes. The highest number of interactions was observed for estrogen related receptor alpha (*ESRRA*) in upregulated DE mRNAs in the LW pigs. For the downregulated DE mRNAs in LW pigs, the highest number of interactions was observed for versican (*VCAN*) and proenkephalin (*PENK*) (Figure 3A). As is predicted by the ingenuity pathway analysis (IPA), *ESRRA* and *VCAN* were regulated by peroxisome proliferative activated receptor, gamma, coactivator 1 alpha (*PPARGC1A*), whereas FKBP prolyl isomerase 5 (*FKBP5*) was regulated by vascular endothelial growth factor (*VEGF*) family and growth hormone (Figure 3B).

### 3.3. Differentially Expressed miRNAs in the LD Muscle between LW and DLY Pigs

Six upregulated and six downregulated DE miRNAs were identified in the LD muscles of LW pigs compared to DLY pigs (Figure 4A) and were clustered into two clades of LW and DLY pigs, respectively (Figure 4B). The greatest fold change in miRNA expression was *ssc-miR1379-5p* (upregulated) and *ssc-miR204-5p* (downregulated), respectively, and all of the DE miRNAs in porcine LD muscle are shown in Table 5 and Table 6. Predictive analysis using the ncRNA targeting database revealed 341 target genes of DE miRNAs, among which 208 genes were targeted by upregulated miRNAs and 212 genes by downregulated miRNAs, and 79 genes were overlapped. GO and KEGG enrichment analysis on these genes indicated that they were mainly enriched in histone methyltransferase complex (Figure 4C), metabolic pathways (Figure 4E) and ubiquitin mediated proteolysis (Figure 4D), and cAMP, RAP1 and Ras signaling pathways (Figure 4F). All of the DE miRNAs were validated by RT-qPCR, showing a significant correlation of 0.859 (*p* < 0.01) (Figure 4G).

### 3.4. Differentially Expressed lncRNAs in the LD muscle between LW and DLY Pigs

A total of 57 DE lncRNAs, 32 upregulated and 25 downregulated, were identified in the LD muscles of LW pigs compared to DLY pigs (Figure 5A) and were clustered into two clades of LW and DLY pigs, respectively (Figure 5B). The greatest fold change in lncRNA expression was *LOC110257307* (upregulated) and *LOC110259141* (downregulated), respectively; and all of the DE lncRNAs in porcine LD muscle are shown in Table 7 and Table 8. Furthermore, 239 genes targeted in *cis* by DE lncRNAs were predicted, including 139 genes targeted by upregulated DE lncRNAs and 111 genes targeted by downregulated lncRNAs (11 genes were overlapped). A total of 163 genes targeted in *trans* by DE lncRNAs were predicted, including 40 genes targeted by upregulated DE lncRNAs and 123 genes targeted by downregulated DE lncRNAs.

GO and KEGG analysis revealed that defense responses to virus (Figure 5C) and metabolic pathways (Figure 5D) were mainly enriched by *cis* targeted genes of DE lncRNAs upregulated in LW pigs. DBird complex (Figure 5E) and necroptosis and MAPK signaling pathways (Figure 5F) were mainly enriched by *cis* targeted genes of DE lncRNAs downregulated in LW pigs. SH3/SH2 adaptor activity (Figure 5G) and adipocytokine signaling pathway and JAK-STAT signaling pathway (Figure 5H) were mainly enriched by *trans* targeted genes of DE lncRNAs upregulated in LW pigs. Structural molecule activity (Figure 5I) and purine metabolism and endocytosis (Figure 5J) were mainly enriched by *trans* targeted genes of DE lncRNAs downregulated in LW pigs. Ten of the randomly selected DE lncRNAs were validated by RT-qPCR, showing a significant correlation of 0.807 (*p* < 0.01) (Figure 5K).

### 3.5. Integrated Analysis on DE mRNAs, miRNAs, and lncRNAs in the LD Muscle between LW and DLY Pigs

The mRNA-miRNA regulatory networks were constructed according to gene expression patterns in the LD muscle of LW pigs: “mRNA up-miRNA down” and “mRNA down-miRNA up”. Five upregulated mRNA genes, including *GALNT15*, *FKBP5*, *PPARGC1A*, *LOC110258214* and *LOC110258215*, and six downregulated miRNA genes, including *ssc-let-7a*, *ssc-miR190-3p*, *ssc-**miR356-5p*, *ssc-**miR573-5p*, *ssc-miR204-5p*, and *ssc-miR-10383*, formed one regulatory network (Figure 6A,B). Three downregulated DE mRNA genes, including *IFRD1*, *LOC110258600* and *LOC102158401*, and six upregulated DE miRNAs, including *ssc-miR1379-3p*, *ssc-miR1379-5p*, *ssc-miR397-5p*, *ssc-miR1358-5p*, *ssc-miR299-5p* and *ssc-miR1156-5p*, formed another network (Figure 6A,C). The expression changes of these mRNA (Figure 6D) and miRNA genes (Figure 4G) in the LD muscle between LW and DLY pigs were validated by RT-qPCR. Based on the mRNA and ncRNA binding site targeting database, we constructed a regulatory network with miRNA as the center and mRNA and lncRNA as the target genes (Figure 7), forming a ceRNA network in the LD muscles that are differentially expressed between LW and DLY pigs.

### 3.6. The Role of GALNT15 in Lipid Deposition

By prediction, *GALNT15, PPARGC1A* and *FKBP5*, are targets of *ssc-let-7a* (Figure 6B), among which the role of *PPARGC1A* and *FKBP5* in lipid deposition has been confirmed [41,42]. We further tested the role of *GALNT15* in lipid deposition. 3T3-L1 cells transfected with overexpression vector pcDNA3.1(+)-GALNT15 showed increased expression of *GALNT15* (Figure 8A), more cells containing oil droplets and significantly upregulated mRNA expression of adipogenic marker genes *CEBPα* and *FASN* (Figure 8B). Knockdown (KD) of *GALNT15* dramatically downregulated the level of oil droplets in 3T3-L1 cells and the mRNA expression of adipogenic marker genes (*PPARγ, CEBPα, FASN* and *SCD*) (Figure 8C), while overexpression of *GALNT15* in these cells recovered the oil droplet level. (Figure 8D). These results indicated that *GALNT15* plays a positive role in the process of lipid deposition.

## 4. Discussion

Analysis of transcriptome profiling of mRNA and noncoding RNA is more and more widely used as a strategy to investigate the mechanism of IMF deposition and muscle development [43]. IMF is the total lipid associated with all cells present in a meat tissue, including extramyocellular lipids and intramyocellular lipids [44]. We found that, at 150d, the IMF of the *Longissimus dorsi* muscle between LW and DLY pigs was different, and LW pigs had more IMF content and slow myofibers and less fast myofibers in the LD muscle. Studies suggest that the difference in IMF is caused by differences in transcript abundance of genes [45], and there is a significant positive correlation between the content of IMF and the expression of lipid metabolism genes [46]. Due to that IMF deposition greatly differs between fat- and lean-type pig breeds, in this study, to identify other genes especially noncoding RNA genes, we further systematically compared the transcriptomes of mRNA, miRNA and lncRNA in the LD muscle between LW and DLY pigs of 150 d, a time point when intramuscular fat is rapidly deposited in LW pigs [23].

Lean- and fat-type pigs exhibit various differences except IMF, which is caused by genetic background as well as other factors including nutrition and farm management. Omics comparisons between lean- and fat-type pigs were reported in several pig breeds [10,11,12,13,14]. To reveal genes underlying differences in IMF and other meat quality traits between LW and DLY pigs, the individuals of both breeds were sampled from the farms within the same region, and the pigs were reared with the same diet (crude protein 16.5% and digestive energy 13.63 MJ/Kg) and slaughtered at the same date (150 d). To control variations within groups, three individuals with similar live weight and IMF were used for transcriptome analysis. By using these samples with great difference in IMF between LW and DLY pigs, we identified differentially expressed mRNAs, miRNAs and lncRNAs genes that are related to IMF deposition and skeletal muscle development and constructed a regulatory network of these three types of genes, which provides resource data for further elucidating mechanisms underlying meat quality variations between lean- and fat- types of pigs.

Comparison of the mRNA transcriptome changes in the LD muscle between LW and DLY pigs identified 225 DE genes, with *LDHB* showing the greatest upregulation in LW pigs. *LDHB* encodes the B subunit of lactate dehydrogenase enzyme, which catalyzes the interconversion of pyruvate and lactate with concomitant interconversion of NADH and NAD+ in a post-glycolysis process [47]. Consistently, the mRNA expression of *LDHB* is significantly lower in Pietrain pigs than Duroc and Duroc–Pietrain crossbred pigs [48] and *LDHB* expression is positively correlated with IMF in pigs crossed by (Pietrain × Duroc) boars and (Landrace × Yorkshire) sows [49], suggesting an important role of *LDHB* in porcine fat deposition traits. KEGG enrichment analysis showed that the upregulated mRNAs were mainly enriched in metabolic pathways (*PTGES2*, *PNP*, *GALNT15*, *KYAT1*, *NNT*, *LDHB*, *ST8SIA5*, *GOT1*, *LOC100525112*, *LOC100737183*, *LOC100739101*, *LOC110255237*) and adipocytokine signaling pathways (*PPARGC1A*). GO enrichment analysis showed that the upregulated mRNAs were mainly enriched in glycerol biosynthetic process (*GOT1*) and fatty acid homeostasis (*GOT1*). Adipocytokine signaling is a crucial pathway for IMF deposition and lipid metabolism [50]. *GOT1* (glutamic-oxaloacetic transaminase 1)-related pathways are glucose metabolism and metabolism. One recent study showed that *GOT1* regulates adipocyte differentiation by altering NADPH content [51]. A recent study showed that *PTGES2* (Prostaglandin E synthase 2) is essential for effective skeletal muscle stem cell function, augmenting regeneration and strength [52]. *NNT* (mitochondrial nicotinamide nucleotide hydrogenase) is the major enzyme that generates NADH in mitochondria, and catalyzes the transfer of a hydride between NADH and NADP+ [53]. KEGG enrichment analysis showed that the downregulated mRNA genes were mainly enriched in metabolic pathways (*PLA2G4E*, *PLPP1*, *TKTL2*, *DHCR24*, *MTHFD1*, *AK5*, *LOC106505238*) and the MAPK signaling pathway (*PLA2G4E*, *FLNC*, *HSPA2*, *PFN2*, *HSP70.2*, *HSPA6*, *IGF2*). It has been reported that the MAPK pathway is the main regulator of skeletal muscle development [54]. *PLA2G4E* is one of the important members of the PLA2 family, and it regulates skeletal muscle and metabolic diseases [55]. *AK5* (adenylate kinase 5) and AMP signaling are necessary for energy communication between mitochondria, myofibrils and nuclei, as well as metabolic programs that promote cardiac differentiation in stem cells [56]. GO enrichment analysis showed that the downregulated mRNA genes were mainly enriched in actin binding (*XIRP1*, *SLC6A2*, *FLNC*, *ENAH*, *MYH15, DBN1*, *PFN2*, *CNN1*, *ANKRD1*, *DNASE1*), consistent with a report that actin participates in maintaining muscle function and ensuring muscle contraction [57].

To better understand the potential relationship between DE genes, we subsequently performed PPI network analysis and revealed that the most frequently involved gene in the interaction was *ESRRA* that was upregulated in LW pigs. *ESRRA* acts as a site-specific transcription factor and interacts with members of the PGC-1 family of transcription cofactors to regulate the expression of most genes involved in cellular energy production as well as mitochondrial biogenesis [58]. By co-activating with *ESRRA*, *PPARGC1A* promotes *LDHB* transcription and regulates skeletal muscle metabolism [59]. *PPARGC1A* also enhances lipid oxidation to provide energy for sustained muscle contraction and regulates glucose metabolism [60]. In this study, the mRNA expression of *PPARGC1A* in the LD muscle was higher in LW pigs compared to DLY pigs, suggesting a regulatory role of *PPARGC1A* in skeletal muscle development. In the gene network constructed by downregulated genes in LW pigs, *VCAN* and *PENK* were most frequently involved in the interaction. *VCAN* was involved in cell adhesion, proliferation, migration and angiogenesis [61,62]. *PENK* encodes small endogenous opioid peptides and plays a critical role in cell proliferation and differentiation [63]. As an endoplasmic reticulum localized protein, *SHISA2* regulates the fusion of muscle satellite cell-derived primary myoblasts [64]. Furthermore, VEGF factors and growth hormone were predicted as upstream regulators of *FKBP5*, which is closely related to the growth and development of skeletal muscle cells or tissues. In 3T3-L1 cells, overexpression of *FKBP5* promotes lipid deposition [42]. In pigs, the expression of *FKBP5* is responsive to glucocorticoid receptor NR3C1 [65,66]; it is involved in activating T lymphocyte [67] and significantly contributes to the breeding value for residual feed intake [68]. Their roles in porcine skeletal muscle development require further study.

The regulation of miRNA in a variety of biological processes has attracted more attention. KEGG enrichment analysis showed that the target genes of downregulated miRNA were enriched in metabolic pathways (*IMPDH1*, *GNE*, *HGSNAT*, *SAT1*, *MTMR8*, *NT5C3A*, *PANK2*, *PHYKPL*, *GALNT11*) and the PPAR signaling pathway (*ANGPTL4*). The PPAR pathway is closely related to glucose homeostasis and lipid metabolism [69]. *ANGPTL4* (angiopoietin-like protein family 4) has been shown to regulate lipoprotein metabolism through the inhibition of lipoprotein lipase (*LPL*) [70]. GO enrichment analysis showed that the target genes of downregulated miRNA included adipose tissue development (*SLC25A25*). *SLC25A25* is reported to regulate lipid metabolism by *PGC1-α* [71]. KEGG enrichment analysis showed that the target genes of upregulated miRNA were enriched in the cAMP pathway (*GNAS*, *GLI3*, *AKT1*, *LOC106505329*) and Ras signaling pathways (*AKT1*, *RALA*, *BCL2L1*). The cAMP signaling pathway plays an important role in regulating muscle development and skeletal muscle differentiation [72]. Activation of the PI3K/protein kinase B (Akt) pathway by Ras induces muscle growth but does not alter fiber-type distribution [73]. Studies have shown that *AKT1* and *AKT2* are indispensable for the regulation of preadipocyte and adipocyte number [74]. GO enrichment analysis showed that the target genes of upregulated miRNA were enriched in histone methyltransferase complex (*KDM6A*, *ZFP64*, *NCOA6*, *KMT2C*, *ZNF335*, *LOC100626655*). Histone methyltransferase complexes are widely involved in the regulation of muscle development [75,76]. Mutations in *KDM6A* and *KMT2D* can reduce myocyte differentiation in vitro and damage muscle fiber regeneration in vivo [77].

Integrated analysis of DE mRNA and DE miRNA revealed a targeted relationship among five upregulated DE mRNAs and six downregulated DE miRNAs, of which *ssc-let-7a* is the most abundant and stable miRNA in porcine muscle development [78,79] and it was predicted to target *FKBP5*, *PPARGC1A* and *GALNT15* genes. Among these genes, an increase in *GLANT15* expression was reported in adipose-derived stromal cells through a differentiation period of 21 d [80]. In this study, we tested the role of *GALNT15* in 3T3-L1 cells and found that it played a positive role in lipid deposition, which is consistent with the higher expression of *GALNT15* in LW pigs. Among these miRNAs, loss of miR-204 increases insulin secretion and regulates lipid metabolism [81], while downregulated miR-204 promotes skeletal muscle regeneration [82]. However, further studies are needed to verify the role of *ssc-miR204-5p* in porcine IMF deposition and skeletal muscle development. Three downregulated DE mRNAs and six upregulated DE miRNAs constitute another target relationship, of which *IFRD1* was the common target of both *ssc-miR1379-5p* and *ssc-miR1379-3p*. It has been reported that *IFRD1* can stimulate skeletal muscle regeneration and, as a regulator of MyoD and NF-κB, participate in myoblast differentiation [83], suggesting a possible role in skeletal muscle growth.

LncRNA could participate in a variety of biological processes through different mechanisms [84,85,86]. In this study, we analyzed the interaction between lncRNA and mRNA in *cis*- and *trans*-interactions. In GO and KEGG analysis, the enrichment of *cis*-target genes of DE lncRNA mainly includes metabolic pathways (*NT5M*, *OPLAH*, *ME3*, *GLUL*, *ACO2*, *PEMT*, *LOC100525869*, *LOC110259864*, *PLA2G4E*, *PLA2G4D*, *AMY2*, *LOC100624333*), the PI3K-Akt signaling pathway (*ITGB6*) and the MAPK signaling pathway (*PLA2G4E*, *PLA2G4D*, *TNFRSF1A*). The metabolic pathways mainly include glycerol and phospholipid metabolism. PI3K-Akt and MAPK pathways have been shown to be involved in mediating muscle fiber types and glucose metabolism [87,88]. Some *trans*-target genes of lncRNA are enriched in the AMPK signaling pathway (*IRS1*, *LEPR*) and adipocytokine signaling pathway (*IRS1*, *LEPR*), which have been shown to be involved in the regulation of lipid deposition and muscle development [89], suggesting a possible role of these DE lncRNA in regulating porcine meat quality traits. *IRS1* has been shown to be necessary for myoblast differentiation and glucose metabolism [90]. *LEPR* (leptin receptor) mutations have been shown to be associated with lipid metabolism [91]. In this study, we constructed an mRNA–miRNA–lncRNA regulatory network and found potential pathways that may regulate intramuscular fat deposition and muscle development in pigs. Similarly, a comparison of the expression profiling of mRNAs, lncRNAs and circRNAs in the LD muscle of Beijing Black and Yorkshire pigs at 210 d identified DE mRNAs that are mainly enriched in the ECM–receptor interaction, focal adhesion, AMPK signaling pathway, PI3K-Akt signaling pathway, adipocytokine signaling pathway, fatty acid metabolism, and PPAR signaling pathway [92].

Although some genes and pathways identified by this study, such as *FKBP5,* PI3K-Akt signaling and adipocytokine signaling pathways, were also reported in other pig breeds, still a lot of other genes of mRNA, miRNA and lncRNA expressed in the LD muscle were only identified by comparison between LW and DLY pigs. This is likely caused by specific differences in LW pigs that have super capability of fat deposition and in the age of pigs used for comparison. Whether these genes play similar roles in IMF deposition and other meat quality traits needs further investigations with more individuals of different pig breeds.

## 5. Conclusions

In this study, we performed a comparative transcriptome analysis of mRNA, miRNA and lncRNA in the *Longissimus dorsi* muscle between lean- and fat-type pigs, and found that some upregulated DE mRNAs including *LDHB*, *GALNT15*, *FKBP5*, *PPARGC1A*, *ESRRA*, and their interacting DE miRNAs and lncRNAs were associated with intramuscular fat deposition, and some downregulated DE mRNAs, including *IFRD1*, *VCAN*, *PENK*, *LOC110258600*, *LOC102158401*, and their interacting DE miRNAs and lncRNAs were associated with skeletal muscle development. It is necessary to elucidate the role of *ESRRA* and *PPARGC1A* in regulating *LDHB* transcription in porcine skeletal muscle metabolism and intramuscular fat deposition using more samples and pig breeds. The miRNA *ssc-let-7a* may regulate the expression of *GALNT15*, *PPARGC1A* and *FKBP5* to affect fat deposition in pigs, which requires further investigations using 3T3-L1 and porcine preadipocytes. Furthermore, it is essential to uncover the regulatory mechanism of these genes and to identify variations in important *cis* DNA elements to elucidate the differences in meat quality traits between lean- and fat- types of pigs. The results of this study are helpful for the identification of genes underlying IMF variations in pigs.

## Figures and Tables

**Figure 1 biomolecules-12-01294-f001:**
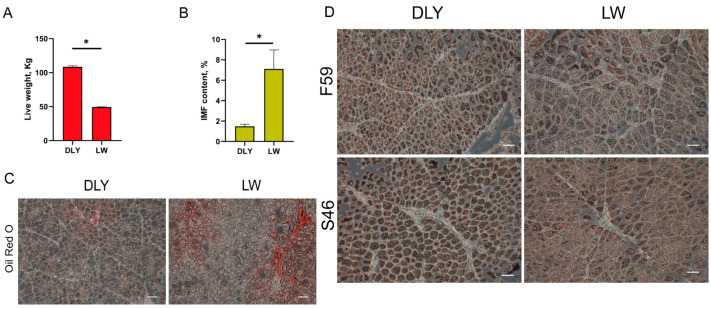
Sample description of the *Longissimus dorsi* muscle from Duroc × Landrace × Yorkshire (DLY) and Laiwu (LW) pigs. Differences in the live weight (**A**) and intramuscular fat content (IMF) (**B**) in the *Longissimus dorsi* (LD) muscles between LW and DLY pigs. * *p* < 0.05. (**C**) Representative images of ORO staining in *Longissimus dorsi* of DLY and LW pigs (Scale bars, 100 μm). *n* = 3 biological samples. (**D**) Immunohistochemical localization of F59 and S46 in *Longissimus dorsi* of DLY and LW pigs (Scale bars, 100 μm). *n* = 3 biological samples.

**Figure 2 biomolecules-12-01294-f002:**
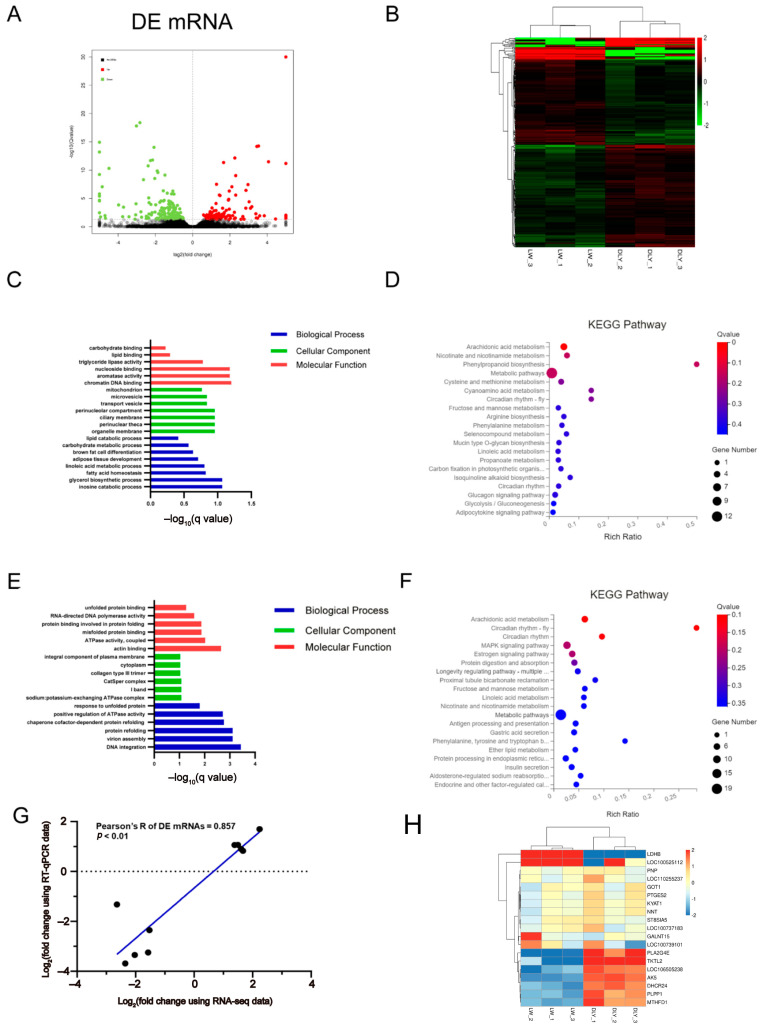
Expression profiles and bioinformatic analysis of DE mRNAs. (**A**) The volcano plots of the significantly differentially expressed mRNAs at the criteria of |log_2_(foldchange)| > 1 and *q* value < 0.05. (**B**) Heatmap plots of the significantly differentially expressed mRNAs. GO (**C**) and KEGG (**D**) analysis diagram of DE mRNAs upregulated in LW pigs. GO (**E**) and KEGG (**F**) analysis diagram of DE mRNAs downregulated in LW pigs. (**G**) Validation by RT-qPCR of 10 randomly selected DE mRNAs from RNA-seq. (**H**) Heatmap plots of the DE mRNAs in metabolic pathway.

**Figure 3 biomolecules-12-01294-f003:**
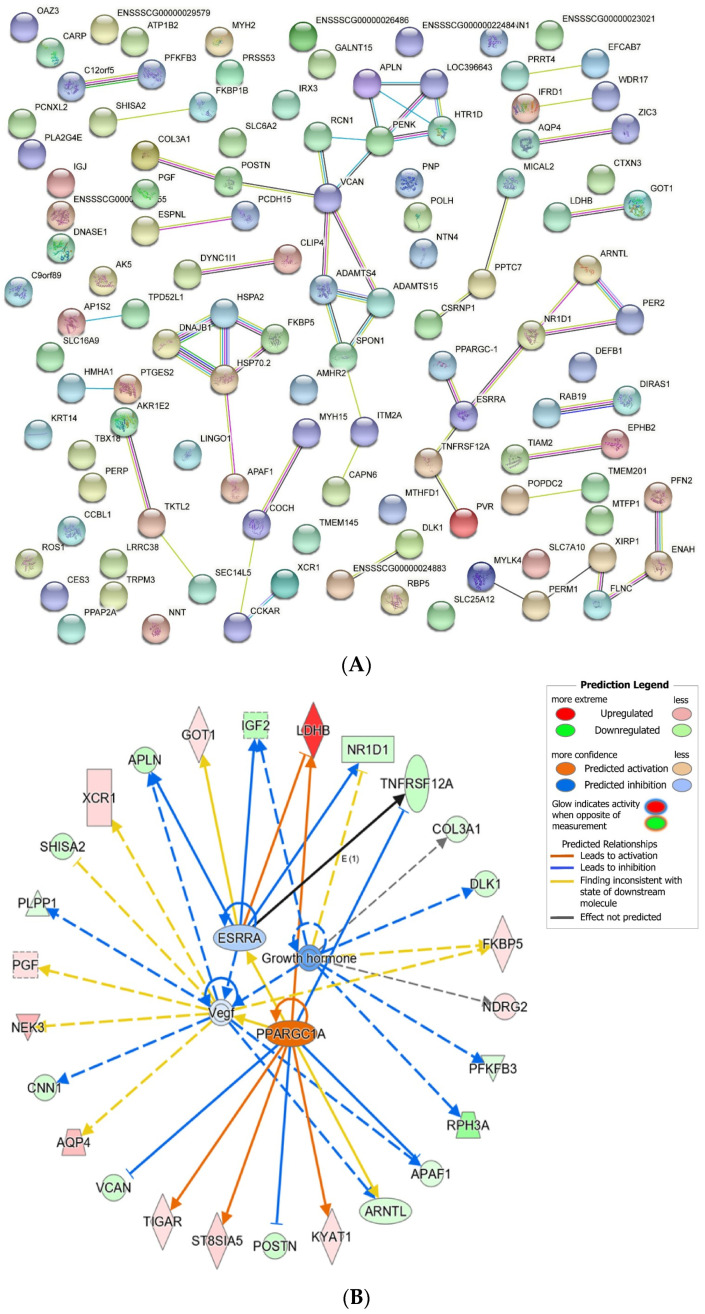
Interaction network for DE mRNAs. (**A**) Protein-protein interaction of DE mRNAs. The network nodes are proteins and the edges represent the predicted functional associations. (**B**) The gene interaction network of DE mRNAs by the ingenuity pathway analysis (IPA). The solid line represents direct interaction; dotted line represents indirect interaction.

**Figure 4 biomolecules-12-01294-f004:**
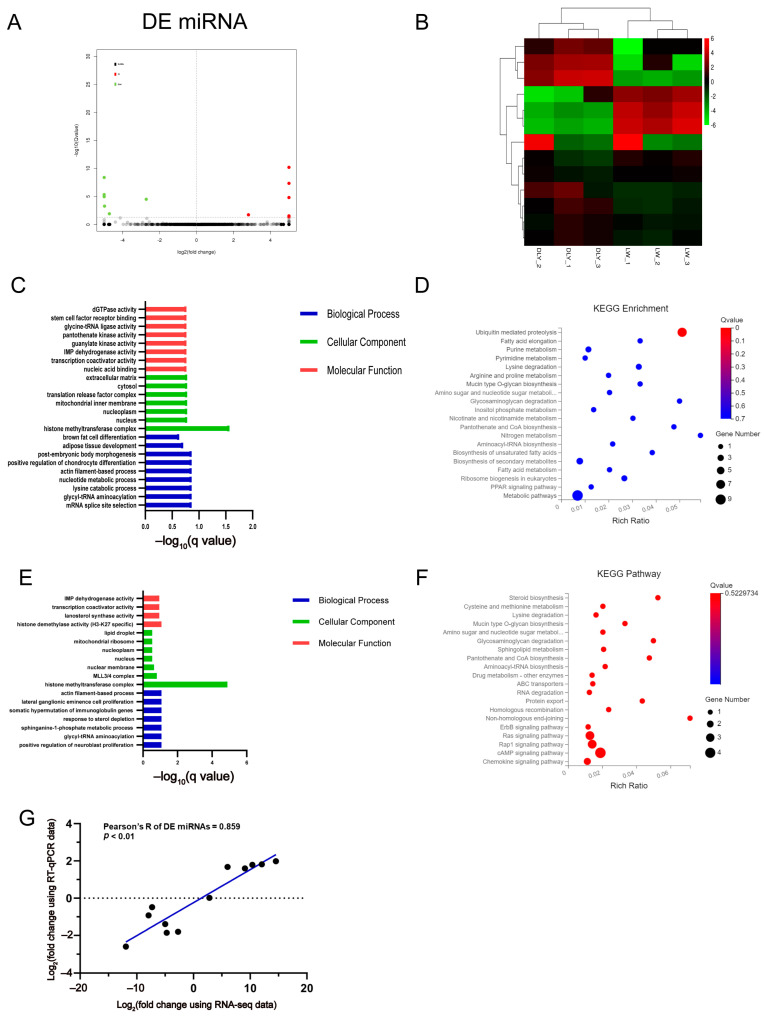
Expression profiles of DE miRNAs and bioinformatic analysis of their targeted genes. (**A**) The volcano plots of the significantly differentially expressed miRNAs at the criteria of |log_2_(foldchange)| > 1 and *q* value < 0.05. (**B**) Heatmap plots of the significantly differentially expressed miRNAs. GO (**C**) and KEGG (**D**) analysis diagram of the target genes of DE miRNAs downregulated in LW pigs. GO (**E**) and KEGG (**F**) analysis diagram of the target genes of DE miRNAs upregulated in LW pigs. (**G**) Validation by RT-qPCR of 12 DE miRNAs from RNA-seq.

**Figure 5 biomolecules-12-01294-f005:**
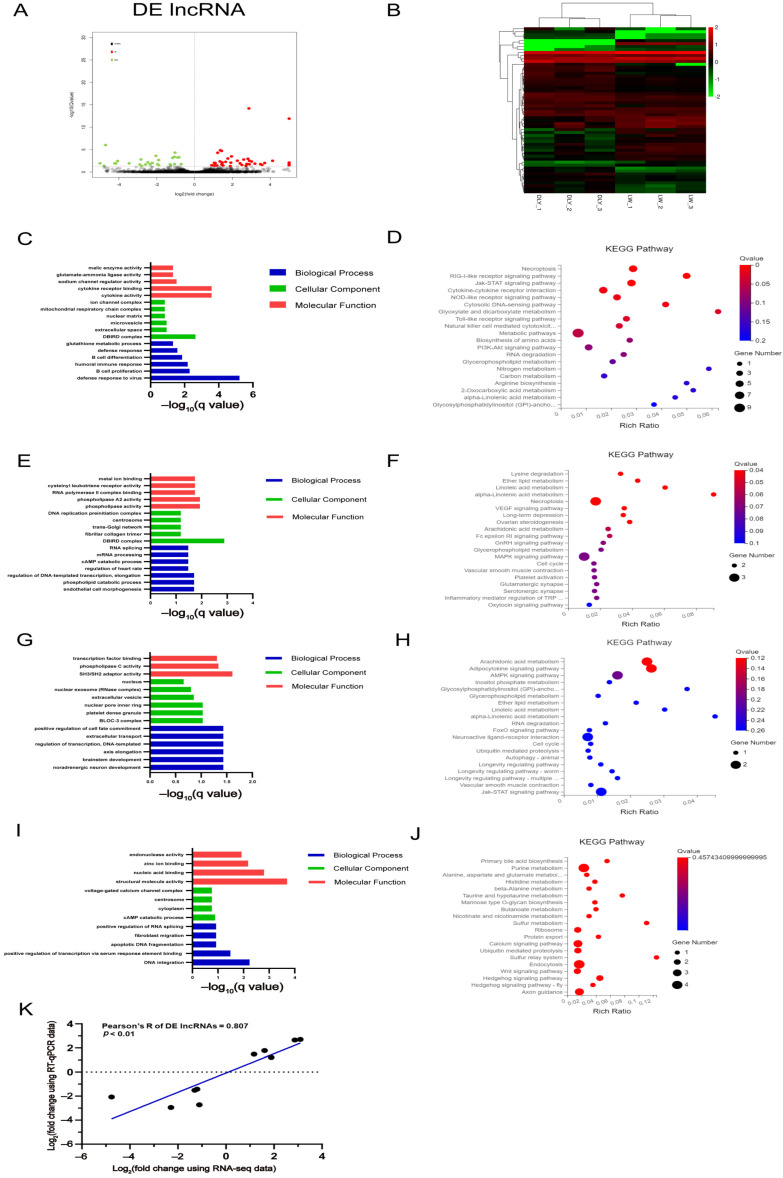
Expression profiles of DE lncRNAs and bioinformatic analysis of its targeted genes. (**A**) The volcano plots of the significantly differentially expressed lncRNAs at the criteria of |log_2_(foldchange)| > 1 and *q* value < 0.05. (**B**) Heatmap plots of the significantly differentially expressed lncRNAs. GO (**C**) and KEGG (**D**) analysis diagram of the *cis* target genes of DE lncRNAs upregulated in LW pigs. GO (**E**) and KEGG (**F**) analysis diagram of the *cis* target genes of DE lncRNAs downregulated in LW pigs. GO (**G**) and KEGG (**H**) analysis diagram of the *trans* target genes of DE lncRNAs upregulated in LW pigs. GO (**I**) and KEGG (**J**) analysis diagram of the *trans* target genes of DE lncRNAs downregulated in LW pigs. (**K**) Validation by RT-qPCR of 10 DE lncRNAs from RNA-seq.

**Figure 6 biomolecules-12-01294-f006:**
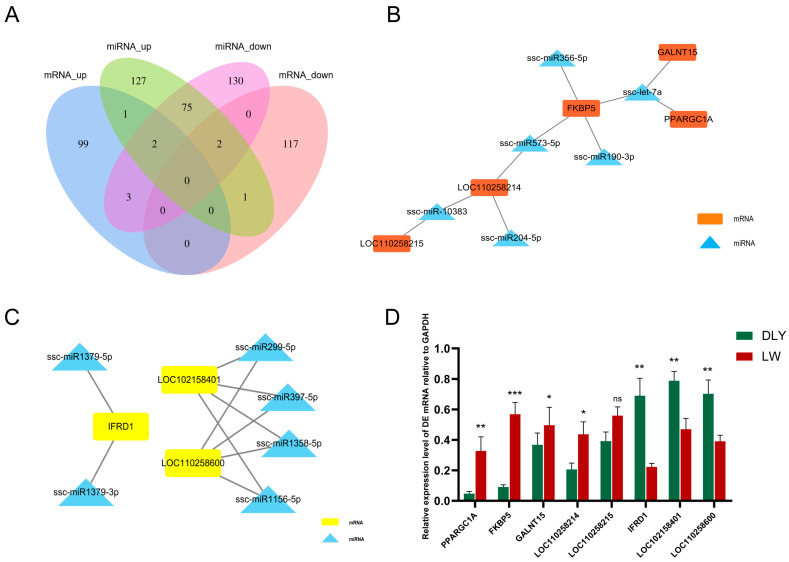
mRNA-miRNA network constructed with DE miRNAs and DE mRNAs in the *Longissimus dorsi* muscle of pigs. (**A**) Venn plots showing the number of overlapping targeted genes of DE miRNAs and DE mRNAs. (**B**) Cytoscape showing the interactions of upregulated DE mRNA and downregulated DE miRNA, five mRNAs and six miRNAs are targeted in this sub-network. The red and blue nodes represent mRNAs and miRNAs, respectively. (**C**) Cytoscape showing the interactions of downregulated DE mRNA and upregulated of DE miRNA; three mRNAs and six miRNAs are targeted in this sub-network. The yellow and blue nodes represent mRNAs and miRNAs, respectively. (**D**) Verification of the above DE mRNAs. * *p* < 0.05, ** *p* < 0.01, *** *p* < 0.001; ns, not significant.

**Figure 7 biomolecules-12-01294-f007:**
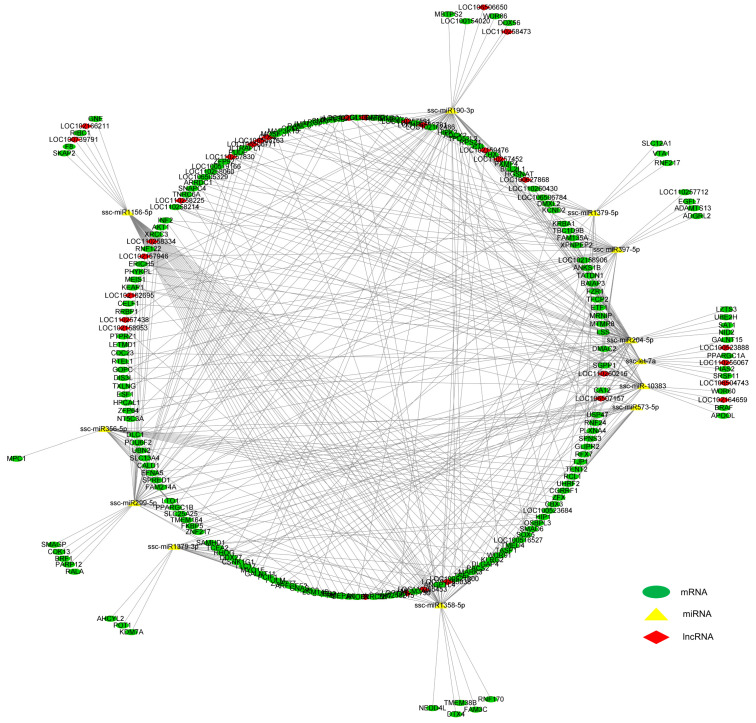
mRNA–miRNA–lncRNA regulatory networks in the *Longissimus dorsi* muscle of pigs. Green circulars indicate mRNAs, red diamonds indicate lncRNAs, yellow triangles indicate miRNAs, and the lines represent the targeting interactions between them.

**Figure 8 biomolecules-12-01294-f008:**
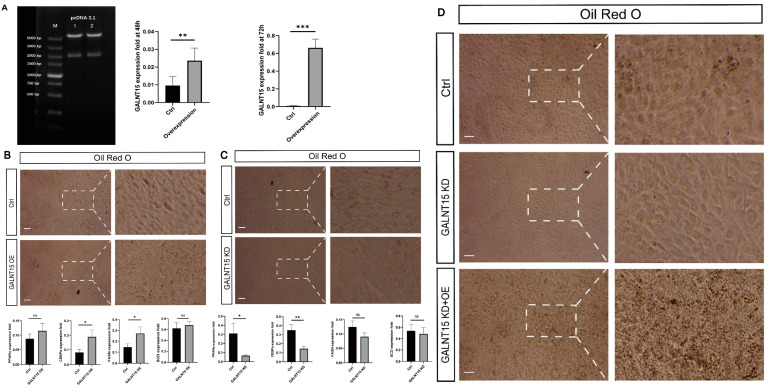
*GALNT15* promotes adipogenesis in 3T3-L1 cells. (**A**) Double enzyme digestion verification of *GALNT15* overexpression plasmid. M: DNA marker. Lines 1 and 2: two replicates of pcDNA3.1(+)-GALNT15 plasmid. (**B**) Representative images of ORO staining in 3T3-L1 cells treated with control (Ctrl) or GALNT15 OE for 48 h (Scale bars, 100 μm). *n* = 3 biological samples. RT-qPCR analysis of GALNT15 OE for adipogenic marker genes. (**C**) Representative images of ORO staining in 3T3-L1 cells treated with Ctrl or GALNT15 KD for 48 h (Scale bars, 100 μm). *n* = 3 biological samples. RT-qPCR analysis of GALNT15 KD for adipogenic marker genes. (**D**) Representative images of ORO staining in 3T3-L1 cells treated with Ctrl, GALNT15 KD and GALNT15 KD+OE. *n* = 3 biological samples. Error Bar indicated SEM, * *p* < 0.05, ** *p* < 0.01, *** *p* < 0.001. ns, not significant.

**Table 1 biomolecules-12-01294-t001:** Statistics of cDNA libraries of mRNA and lncRNA of *Longissimus dorsi* muscle samples from LW and DLY pigs.

Sample	DLY_1	DLY_2	DLY_3	LW_1	LW_2	LW_3
Total raw reads (M)	122.44	122.44	122.44	124.94	122.44	124.94
Total clean reads (M)	113.94	113.95	113.85	116.11	113.82	116.32
Total clean data (Gb)	11.39	11.40	11.39	11.61	11.38	11.63
Total mapping (%)	93.21	93.10	93.74	92.38	92.61	92.44
Clean reads Q20 (%)	97.50	97.47	97.37	97.45	97.38	97.50
Clean reads Q30 (%)	91.24	91.17	90.68	91.00	90.70	91.06
Clean reads ratio (%)	93.06	93.07	92.98	92.93	92.96	93.11

**Table 2 biomolecules-12-01294-t002:** Statistics of cDNA libraries of miRNA of *Longissimus dorsi* muscle samples from LW and DLY pigs.

Sample	DLY_1	DLY_2	DLY_3	LW_1	LW_2	LW_3
Raw Tag Count	25,165,824	25,165,824	25,165,824	25,165,824	25,165,824	25,165,824
Clean Tag Count	23,786,351	22,838,322	24,113,825	24,039,051	24,316,156	24,243,603
Total mapping (%)	90.71	93.45	92.13	91.30	89.07	91.26
Q20 of Clean Tag (%)	98.70	98.60	98.60	98.50	98.70	98.70
Percentage of Clean Tag (%)	94.52	90.75	95.82	95.52	96.62	96.34

**Table 3 biomolecules-12-01294-t003:** The upregulated mRNAs in the *Longissimus dorsi* muscle sample of LW pigs according to *q* value threshold of <0.05 and log_2_ (foldchange) > 3.

Gene ID	Genes	Log_2_ (Foldchange)	*q* Value	DLY Average Read Counts	LW Average Read Counts
100621540	LDHB	9.667279398	6.38 × 10^−12^	0.19	158.17
110258214	LOC110258214	6.986698187	5.01 × 10^−167^	391.55	49,658.91
396643	SSTR3	6.043282076	0.010101424	0.19	12.68
110255206	LOC110255206	5.935360726	0.015904385	0.19	11.72
100737113	LOC100737113	5.856804446	0.015236088	0.72	41.86
100737631	LOC100737631	5.78009055	0.015904385	0.19	10.51
106506226	LOC106506226	5.265941864	0.031736179	9.43	362.74
102159048	SYCP2L	5.181170867	0.029172368	0.65	23.6
100514305	STPG2	5.095150238	0.031736179	0.38	12.88
110256818	LOC110256818	4.451349764	0.045998639	0.71	15.63
100158003	LOC100158003	4.070632429	3.88 × 10^−12^	146.32	2458.62
100525112	LOC100525112	3.847920892	0.013043917	1.94	27.88
110255992	SPATA32	3.596918469	0.01550072	3.07	37.2
102162178	LOC102162178	3.538200994	2.77 × 10^−15^	11	127.83
100514435	RAB19	3.482899896	0.036286962	2.17	24.23
100513679	NEK3	3.447440519	6.95 × 10^−15^	22.96	250.51
100155596	HRK	3.420051236	0.029848272	1.68	17.98
100513483	DCAF16	3.395259004	0.005534146	2.45	25.75
100156672	NEK5	3.178863345	3.65 × 10^−4^	10.29	93.2
100156863	ZIC3	3.11041838	6.68 × 10^−4^	12.58	108.64
100126277	AQP4	3.030803942	4.32 × 10^−5^	118.82	971.11

**Table 4 biomolecules-12-01294-t004:** The downregulated mRNAs in the *Longissimus dorsi* muscle sample of LW pigs according to *q* value threshold of <0.05 and log_2_ (foldchange) < −3.

Gene ID	Genes	Log_2_ (Foldchange)	*q* Value	DLY Average Read Counts	LW Average Read Counts
110258854	LOC110258854	−23.2798	1.28 × 10^−6^	108.47	0
106510102	LOC106510102	−7.50914	5.68 × 10^−14^	172.7	0.95
100737845	FBXL13	−7.44403	3.85 × 10^−6^	59.49	0.34
106510322	LOC106510322	−7.22045	2.85 × 10^−5^	25.85	0.17
100517716	SLC6A2	−6.05305	2.91 × 10^−5^	82.52	1.24
110257759	LOC110257759	−5.92607	0.003691	20.57	0.34
100626756	ACBD7	−5.25986	9.63 × 10^−10^	158.71	4.14
100157711	LOC100157711	−5.16206	2.77 × 10^−15^	533.3	14.89
100624179	COCH	−4.93751	0.020584	28.48	0.93
397171	OCA2	−4.84709	1.48 × 10^−7^	183.52	6.38
100621079	MYH15	−4.69801	0.012088	33.34	1.28
100518260	UNC79	−4.67715	0.038978	63.76	2.49
106505804	LOC106505804	−4.48474	1.50 × 10^−10^	146.11	6.53
100157276	ADAMTS4	−3.96958	1.96 × 10^−4^	188.52	12.03
100623128	CCKAR	−3.75171	0.017717	17.12	1.27
100514811	PLA2G4E	−3.70605	4.49 × 10^−4^	91.7	7.03
100736858	RPH3A	−3.41325	0.017239	16.88	1.58
110256933	LOC110256933	−3.05582	0.009215	39.66	4.77
100620172	IRX5	−3.03576	1.13 × 10^−4^	85.81	10.46
110256218	LOC110256218	−3.01193	1.96 × 10^−18^	740.73	91.83

**Table 5 biomolecules-12-01294-t005:** The upregulated miRNAs in the *Longissimus dorsi* muscle of LW pigs.

Genes	Log_2_ (Foldchange)	*q* Value	DLY Average Read Counts	LW Average Read Counts
ssc-miR1379-5p	14.53752176	6.38 × 10^−11^	0.19	286.08
ssc-miR1379-3p	12.04916787	1.55 × 10^−5^	0.19	50.99
ssc-miR1156-5p	10.38801729	0.045224024	0.19	15.71
ssc-miR1358-5p	9.081557409	4.56 × 10^−8^	0.37	175.8
ssc-miR397-5p	6.009154715	0.029208088	0.37	20.77
ssc-miR299-5p	2.808214579	0.018798556	136.45	869.54

**Table 6 biomolecules-12-01294-t006:** The downregulated miRNAs in the *Longissimus dorsi* muscle sample of LW pigs.

Genes	Log_2_ (Foldchange)	*q* Value	DLY Average Read Counts	LW Average Read Counts
ssc-miR204-5p	−11.91438513	4.70 × 10^−6^	50.8	0
ssc-let-7a	−7.912724505	4.10 × 10^−9^	68,300.09	257.16
ssc-miR190-3p	−7.309917116	1.09 × 10^−5^	100.65	0.61
ssc-miR573-5p	−4.991746779	5.32 × 10^−4^	52.53	1.53
ssc-miR356-5p	−4.719263592	0.012202205	42.26	1.5
ssc-miR-10383	−2.723136638	1.40 × 10^−7^	7773.92	1083.98

**Table 7 biomolecules-12-01294-t007:** The upregulated lncRNAs in the *Longissimus dorsi* muscle sample of LW pigs.

Gene ID	Genes	Log_2_ (Foldchange)	*q* Value	DLY Average Read Counts	LW Average Read Counts
110257307	LOC110257307	6.435958	0.012209	0.19	16.81
110258489	LOC110258489	6.250708	0.006616	0.19	14.76
110255522	LOC110255522	5.833916	0.015984	0.19	10.96
110259865	LOC110259865	5.703756	0.029037	0.19	10
102167235	LOC102167235	4.085308	0.005753	1.42	24.06
102160723	LOC102160723	3.657797	0.012209	1.65	20.77
110257281	LOC110257281	3.464625	0.012209	3.22	35.57
102159627	LOC102159627	3.459035	0.012206	8.95	98.4
102162221	ZNF648	3.094487	0.006616	16.78	143.37
110259814	LOC110259814	2.94117	0.003298	8.46	64.97
100157061	LOC100157061	2.868918	0.003501	7.94	58.03
110260631	LOC110260631	2.860203	0.012209	7.29	52.92
106509142	LOC106509142	2.824431	1.47 × 10^−12^	32.41	229.59
110257827	LOC110257827	2.748534	0.00103	84.95	570.9
110255857	LOC110255857	2.730341	0.021514	35.06	232.65
110257909	LOC110257909	2.691043	0.00752	13.71	88.55
110259137	LOC110259137	2.648392	0.029867	19.6	122.89
110260388	LOC110260388	2.553643	0.005825	6	35.21
110260634	LOC110260634	2.47726	0.022658	9.95	55.39
110257745	LOC110257745	2.349905	0.015074	6.69	34.11
110256935	LOC110256935	2.211555	0.003278	46.14	213.71
110259691	LOC110259691	1.911688	0.00148	48.79	183.58
106505263	LOC106505263	1.887082	0.014899	19.65	72.68
110255980	LOC110255980	1.606907	0.01395	32.63	99.38
102163278	LOC102163278	1.424182	0.032409	26.94	72.28
110261569	LOC110261569	1.380077	0.032399	60.42	157.25
106507563	LOC106507563	1.30465	0.025425	70.18	173.37
100622439	LOC100622439	1.170522	0.004696	241.41	543.4

**Table 8 biomolecules-12-01294-t008:** The downregulated lncRNAs in the *Longissimus dorsi* muscle sample of LW pigs.

Gene ID	Genes	Log_2_ (Foldchange)	*q* Value	DLY Average Read Counts	LW Average Read Counts
110259141	LOC110259141	−6.21526	0.007991	12.48	0.17
110260256	LOC110260256	−4.7651	4.39 × 10^−7^	65.94	2.43
110260878	LOC110260878	−4.29011	0.012206	17.59	0.9
110256028	LOC110256028	−4.21142	0.002245	44.64	2.41
110260798	LOC110260798	−4.12382	0.00148	48.52	2.78
110261211	LOC110261211	−3.52091	0.011148	20.9	1.82
102159118	LOC102159118	−3.14539	0.032653	48.75	5.51
110261331	LOC110261331	−2.94541	0.012272	33.06	4.29
110260773	LOC110260773	−2.87334	0.001713	59.8	8.16
106510214	LOC106510214	−2.84826	0.027653	35.23	4.89
106506422	LOC106506422	−2.63706	0.011148	62.99	10.13
100627270	LOC100627270	−2.39084	0.014715	43.03	8.2
110257276	LOC110257276	−2.33742	0.030236	80.59	15.95
100623270	LOC100623270	−2.29408	0.009702	36.81	7.51
102165609	LOC102165609	−2.08699	5.73 × 10^−4^	237.9	55.99
110256124	LOC110256124	−1.95087	0.02857	42.02	10.87
106509650	LOC106509650	−1.93849	0.00275	824.96	215.22
106508816	LOC106508816	−1.34055	0.030236	111.44	44
106505720	LOC106505720	−1.30212	0.041647	342.85	139.04
110257277	LOC110257277	−1.20335	0.006588	2752.42	1195.28
110255759	LOC110255759	−1.10927	0.032409	200.91	93.13
110255800	LOC110255800	−1.1087	0.009503	818.25	379.43
102163738	LOC102163738	−1.07791	0.005457	293.6	139.08

## Data Availability

The transcriptome data can be accessed from NCBI (https://www.ncbi.nlm.nih.gov/) with the accession number PRJNA815878.

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
