# Peer review of "Comparative Transcriptomic Analysis of mRNAs, miRNAs and lncRNAs in the *Longissimus dorsi* Muscles between Fat-Type and Lean-Type Pigs"

_biomolecules, 2022, doi:10.3390/biom12091294_

Round 1
Reviewer 1 Report (Previous Reviewer 4)
This manuscript has already been peer-reviewed. All suggestions previously recommended have been accepted by the authors, so it is with great pleasure that I can recommend the acceptance of this paper for publication.
Author Response
Thank you for your comments.
Reviewer 2 Report (New Reviewer)
This manuscript was aim to uncover genetic factors controlling differences related to IMF, and perform RNA-seq analysis on the transcriptomes of the LD muscle of LW and DLY pigs. The authors found certain key genes that might providing resources for further investigation of molecular mechanisms of high quality meat traits. Just one question: Whether the diets of LW and DLY pigs are same or different, especially the dietary protein and energy levels?
Author Response
Thank you for your comment. In this study, the LW and DLY pigs were reared with the same diet (crude protein 16.5% and digestive energy 13.63 MJ/Kg). see lines 464 to 467. The purpose of this design is to eliminate influences caused by different nutrition level.
Reviewer 3 Report (New Reviewer)
The work done by Jiang Zhang and cols. entitled: “Comparative transcriptomic analysis of mRNAs, miRNAs and lncRNAs in the longissimus dorsi muscles between fat-type and lean-type pigs”. In this work the authors developed a comparative analysis of skeletal muscle mRNA, miRNA and lncRNA related with intramuscular fat deposition and fat metabolism in two different strains of pigs, the lean pig, Yorkshire, and the fat-type Laiwu pig. In their work, the authors using a powerful tool of RNAseq described a total of 225 mRNAs, 12miRNAs and 57 lncRNAs as differentially expressed under their statistical analysis. The study is well done; however, it needs a few clarifications.
Comments:
1. My main concern is about ethical statement regarding the sacrifice of pigs to obtain 3 grams of muscle tissue. It would have been more than enough just to take a biopsy from the animal?
2. Please state the meaning of abbreviations before use in the whole manuscript (i.e. DE, circRNA)
3. I assume there is a typo on line 20, please verify
4. It is very unclear how do the authors stand for W1, W2, W3 in the formulas on IMF content.
5. How was the housing condition of the animals? How were they fed, etc?
6. It would be very illustrative if the authors make some mapping of the metabolic pathways of the genes differentially expressed between lean and fat pigs.
7. Discussion is lacking of specific information related to the role of differentially expressed genes between lean and fat pigs
Author Response
1.My main concern is about ethical statement regarding the sacrifice of pigs to obtain 3 grams of muscle tissue. It would have been more than enough just to take a biopsy from the animal?
Reply: Thank you for your attention. We raised and sampled animals according to the methods adopted in previous studies, and added reference. See line 96. Different from mice, at 150 d, the pigs are slaughtered for meat production. Because we only compared the differences in transrriptomes at the fixed time point of 150 days, we did not take a biopsy from pigs.
2.Please state the meaning of abbreviations before use in the whole manuscript (i.e. DE, circRNA)
Reply: Thank you for your comments. We added a description of abbreviations in the article (i.e. DE, circRNA). See lines 147, 73 to 74.
3. I assume there is a typo on line 20, please verify
Reply: Thank you for your comments. We revised the description of this sentence. See lines 20 to 22.
4.It is very unclear how do the authors stand for W1, W2, W3 in the formulas on IMF content.
Reply: Thank you for your attention. We revised the description of this section. See lines 102 to 113. The formula is deleted.
5.How was the housing condition of the animals? How were they fed, etc?
Reply: Thank you for your comments. In the materials and methods section, we added the description of the living conditions of pigs. See lines 91 to 92.
6. It would be very illustrative if the authors make some mapping of the metabolic pathways of the genes differentially expressed between lean and fat pigs.
Reply: Thank you for your attention. We added heatmap statistics of DE mRNAs enriched in metabolic pathway and analyzed related genes in the results section. See lines 300 to 304. and Figure 2H.
7. Discussion is lacking of specific information related to the role of differentially expressed genes between lean and fat pigs
Reply: Thank you for your comment. We added the information of some genes, i.e. LDHB, GOT1, PTGES2, PLA2G4E, AK5, in the discussion. See lines 492 to 496 and 502 to 504.
Reviewer 4 Report (New Reviewer)
Comparative Transcriptomic Analysis of mRNAs, miRNAs and lncRNAs in the longissimus dorsi Muscles between Fat-type and Lean-type Pigs By Zhang et al.
This is a very descriptive and not a novel paper. Similar studies have been done before in the pig and in different model system comparing fat and non-fat tissues. There is no new information we can get out from this paper. The data presented are from sequencing and bioinformatics analysis.
There is no effort to verify some of the target identified, to test in a cell culture experiments that makes this paper very weak.
Author Response
This is a very descriptive and not a novel paper. Similar studies have been done before in the pig and in different model system comparing fat and non-fat tissues. There is no new information we can get out from this paper. The data presented are from sequencing and bioinformatics analysis.
There is no effort to verify some of the target identified, to test in a cell culture experiments that makes this paper very weak.
Reply: Thanks for your comments. To uncover genes underlying meat quality traits, we selected two types of pigs with marked difference in IMF and myofiber traits. The LW pigs have much higher IMF than other pig breeds, which are suitable and specific pig breed for this study.
We supplemented the results of functional verification of the candidate gene GALNT15 to make the paper more convincing. By overexpression and knockdown, we tested the role of GANT15 in 3T3-L1 cells and found that it can stimulate lipid deposition. See lines 184 to 210, 424 to 446, 558 to 560.
Round 2
Reviewer 4 Report (New Reviewer)
The authors have significantly improved the paper. There are lots of important data included in this version of the manuscript to make their points.
This manuscript is a resubmission of an earlier submission. The following is a list of the peer review reports and author responses from that submission.
Round 1
Reviewer 1 Report
The authors report the different transcript abundances (mRNAs, miRNAs, lncRNAs) determined by RNAseq and confirmed by qPCR between muscle samples from two breeds of pigs that differ in muscle-fat ratio, especially intramuscular fat. The study is based on a very small number of samples (3 + 3). The manuscript is clearly worded.
The authors fail to address two important points:
- the cell composition of the tissue samples of the two breeds is probably different (more fat cells in the Laiwu), leading to differences in transcript abundance. The authors should discuss this problem and make it clear that "differentially expressed" does not necessarily mean different gene activities, but is a synonym for "differentially abundant transcripts".
- breed comparisons suffer from the fact that all differences shown may be due to various differences and cannot be directly linked to a single trait. This limitation of breed comparisons should also be discussed. Comparisons of extreme animals within a breed are far better; but usually provide fewer "DEs". Furthermore, both breeds were kept on different farms, so that environmental influences can be confused with breed. The authors list a number of references with similar breed comparisons in the introduction. However, these studies are not referred to in the discussion. Were there not pathways discovered here that were shown in earlier studies? How do the authors interpret this if there is no overlap in pathways? Did the different breeds engage in different pathways to achieve similar phenotypes?
Other issues to be considered in a revision:
To obtain miRNA targets, the author used the miRDB. However, the database does not contain pig sequences. Did the author manually access the pig genome sequence information?
Table 9 could be included in the supplements.
The sequencing data must be made available via public databases.
The conclusion is just a summary
Author Response
The authors report the different transcript abundances (mRNAs, miRNAs, lncRNAs) determined by RNAseq and confirmed by qPCR between muscle samples from two breeds of pigs that differ in muscle-fat ratio, especially intramuscular fat. The study is based on a very small number of samples (3 + 3). The manuscript is clearly worded.
Reply: Thank you for your comment. You are right. Three samples for each group are the minimum requirement of transcriptome comparision. To reduce variations within group, the three samples with similar age, live weight and IMF were used.
The authors fail to address two important points:
- the cell composition of the tissue samples of the two breeds is probably different (more fat cells in the Laiwu), leading to differences in transcript abundance. The authors should discuss this problem and make it clear that "differentially expressed" does not necessarily mean different gene activities, but is a synonym for "differentially abundant transcripts".
Reply: In the Discussion section, we addressed this problem and rewrote the first paragraph. See lines 366 to 372.
- breed comparisons suffer from the fact that all differences shown may be due to various differences and cannot be directly linked to a single trait. This limitation of breed comparisons should also be discussed. Comparisons of extreme animals within a breed are far better; but usually provide fewer "DEs". Furthermore, both breeds were kept on different farms, so that environmental influences can be confused with breed. The authors list a number of references with similar breed comparisons in the introduction. However, these studies are not referred to in the discussion. Were there not pathways discovered here that were shown in earlier studies? How do the authors interpret this if there is no overlap in pathways? Did the different breeds engage in different pathways to achieve similar phenotypes?
Reply: I agree with you on this point. In the Discussion section, we addressed the efforts we did to reduce the effects of other factors except genetic components. See the second paragraph of Discussion section, lines 379 to 392. In this paragraph, we also listed references concerning studies between pig breeds, lines 381 to 382. Moreover, in the last paragraph of Discussion section, we discussed the differences in genes identified between studies, see lines 500 to 507.Other issues to be considered in a revision:
To obtain miRNA targets, the author used the miRDB. However, the database does not contain pig sequences. Did the author manually access the pig genome sequence information?
Reply: Yes. We also compared the conservation of human miRNA sequences and porcine miRNA sequences in addition to using miRDB.
Table 9 could be included in the supplements.
Reply: Thank you for your suggestion. We deleted Table9 and submitted the content as supplement.
The sequencing data must be made available via public databases.
Reply: The sequencing data has been uploaded to the GEO database, but has not been released yet. See lines 220 to 221.
The conclusion is just a summary
Reply: The conclusion section was written again. See lines 508 to 518.

Reviewer 2 Report
The authors of "Comparative Transcriptomic Analysis of mRNAs, miRNAs and 2 lncRNAs in the longissimus dorsi Muscles between Fat-type 3 and Lean-type Pigs" performed a transcriptomic comparison between two pig breeds which different intramuscular fat content: Laiwu pigs (3 samples) and DurocxLandraceXYork-shire pigs (3 samples). The analysis performed in the current study scientifically sounds and the results obtained can be relevant to better understanding the biological differences between high and low IMF content in pigs. However, there are important points that should be clarified and better explained. Please, check my detailed comments below.
Major comments:
Abstract:
On lines 18-21 the authors are describing candidate genes. However, the authors should better describe some results. How much higher was the expression of LDHB? Which this gene is being described individually?
On lines 29-30 the authors should describe the results obtained from the regulatory network analysis.
Introduction:
The authors cited several examples of specific miRNAs identified in other studies. I suggest the authors remove these sentences or move to the discussion section.
Material and Methods:
The material and methods section must be substantially improved. The authors don't describe the methodology used to obtain several of the results presented later in the manuscript. There is no mention about the statistical analysis performed to compare the IMF content and the live weight between the breeds.
The authors should remove the sentence present on lines 133-135 as the next sub-section will describe the alignment step. Additionally, the authors should describe in detail all the bioinformatics pipelines used to identify the lncRNAs and the miRNAs. Additionally, there is no mention of the version of the swine reference genome used for the alignment step. The authors should cite the proper reference for the heatmap package and R, respectively.
The authors introduced in the sub-section 2.5 the "Bioinformatics Analysis". However, the previous sub-section already describes bioinformatics analysis. On line 156, which kind of multiple testing correction was applied to the p-values?
The authors should clearly cite the RNAs selected to be validated by qPCR. Why those genes were chosen?
Regarding the "Statistical analysis" sub-section, where is the analysis of IMF comparison? The authors should present the exact number of independent experiments.
Results:
The dataset PRJNA815878 is not available on GEO.
There is no description of the analysis performed to obtain the results presented in Figure 1 in the material and methods section.
The authors should present the percentage of mapped reads for each sample, as well as mean and SD.
Why the top 35 up and downregulated genes were presented? Is there any specific reason for this number?
On lines 230-231 the authors informed that the mRNAs upregulated in the LD muscle of LW pigs are enriched for several GO terms. However, genes/mRNAs/proteins/etc. are not enriched in this kind of analysis. The terms are enriched.
The protein-protein interaction results are not mentioned in the material and methods section.
The authors did not describe the comparison between the miRNAs and the target mRNAs presented on lines 339-340 in the material and methods section. The selected miRNAs and the mRNAs used for this analysis should be clearly shown and described in the material and methods section. The reason why these RNAs were selected should be clearly stated as well.
Discussion:
In general, the discussion is short and doesn't have enough focus on the functional candidate mRNAs, miRNAs, and lncRNAs. Additionally, a better link between the candidate molecules, the enriched biological processes, and the phenotypic differences between the breeds should be presented. How the results obtained help to better understand the differences of IMF between the breeds?
On lines 361-363, the authors cited a previous study from the research group where the transcriptome of LD of LW was analyzed from 60d to 400d. The authors should compare the results obtained in the current study with the results obtained in the previous study. This approach should help to better identify functional candidate genes.
From lines 381-389 the authors compared the number of DEGs from different studies. I suggest the authors remove this comparison and focus more on the candidate genes and networks obtained in the current study.
Conclusion:
The conclusion is poorly written. The authors should use the conclusion to describe how the obtained results fill up the current gap in the literature regarding the knowledge about the biological basis of IMF content in pigs. Additionally, perspectives and limitations could be cited.
Minor comments:
Line 32: "invaluable results". This is speculative.
Lines 52-53: Insert space before reference square brackets.
Lines 60-61: The authors mentioned "several studies" but only two were cited.
Line 132: Introduce space between by and RT-PCR.
Line 162: Introduce space between RNAplex and the citation.
Line 301-308. The sentence is very long.
Table 9 should be moved to Supplementary material.
Line 377: Include reference about NAD+.
Line 380: Introduce space between sows and the reference.
Line 399: Avoid using "much higher" and show the quantitative value.
Line 405: Reference.
Lines 434-435: Which trans-target-genes?
Author Response
The authors of "Comparative Transcript.omic Analysis of mRNAs, miRNAs and 2 lncRNAs in the longissimus dorsi Muscles between Fat-type 3 and Lean-type Pigs" performed a transcriptomic comparison between two pig breeds which different intramuscular fat content: Laiwu pigs (3 samples) and DurocxLandraceXYork-shire pigs (3 samples). The analysis performed in the current study scientifically sounds and the results obtained can be relevant to better understanding the biological differences between high and low IMF content in pigs. However, there are important points that should be clarified and better explained. Please, check my detailed comments below.
Reply: Thank you for your comments.
Major comments:
Abstract:
On lines 18-21 the authors are describing candidate genes. However, the authors should better describe some results. How much higher was the expression of LDHB? Which this gene is being described individually?
Reply: Thank you for your suggestion. We modified the description, See lines 18 to 20.
On lines 29-30 the authors should describe the results obtained from the regulatory network analysis.
Reply: Thank you for your suggestion. We modified the description, See lines 29 to 33.
Introduction:
The authors cited several examples of specific miRNAs identified in other studies. I suggest the authors remove these sentences or move to the discussion section.
Reply: Thank you for your suggestion. We deleted these sentences.
Material and Methods:
The material and methods section must be substantially improved. The authors don't describe the methodology used to obtain several of the results presented later in the manuscript. There is no mention about the statistical analysis performed to compare the IMF content and the live weight between the breeds.
Reply: Thank you for your suggestion. We modified the description and described the statistical method on IMF and live weight. See lines 195 to 196.
The authors should remove the sentence present on lines 133-135 as the next sub-section will describe the alignment step.
Reply: Thank you for your suggestion. We deleted these sentences. The bioinformatic analysis subsection was rewritten. See lines 132 to 167.
Additionally, the authors should describe in detail all the bioinformatics pipelines used to identify the lncRNAs and the miRNAs. Additionally, there is no mention of the version of the swine reference genome used for the alignment step. The authors should cite the proper reference for the heatmap package and R, respectively.
Reply: Thank you for your suggestion. The bioinformatic analysis subsection was rewritten. See lines 155 to 157
The authors introduced in the sub-section 2.5 the "Bioinformatics Analysis". However, the previous sub-section already describes bioinformatics analysis. On line 156, which kind of multiple testing correction was applied to the p-values?
Reply: Multiple testing correction of P-value was applied using R package: https://bioconductor.org/packages/release/bioc/html/qvalue.html. See lines 151 to 153.
The authors should clearly cite the RNAs selected to be validated by qPCR. Why those genes were chosen?
Reply: The DE mRNA genes and lncRNA were randomly selected for RT-qPCR. See lines 246 to 247 and lines 318 to 319. All of the 12 miRNA genes were validated.
Regarding the "Statistical analysis" sub-section, where is the analysis of IMF comparison? The authors should present the exact number of independent experiments.
Reply: We added this statistic. See lines 195 to 196.
Results:
The dataset PRJNA815878 is not available on GEO.
Reply: The sequencing data has been uploaded to the GEO database, but has not been released yet.
There is no description of the analysis performed to obtain the results presented in Figure 1 in the material and methods section.
Reply: We added this statistic. See lines 195 to 196.
The authors should present the percentage of mapped reads for each sample, as well as mean and SD.
Reply: We added this statistic. See lines 214 to 215, 217 to 218.
Why the top 35 up and downregulated genes were presented? Is there any specific reason for this number?
Reply: Generally top 10 was listed. Some study listed top 50. Top 35 is not strictly and specificlly selected. The reason for this selection is to show mor information.
On lines 230-231 the authors informed that the mRNAs upregulated in the LD muscle of LW pigs are enriched for several GO terms. However, genes/mRNAs/proteins/etc. are not enriched in this kind of analysis. The terms are enriched.
Reply: We only listed some terms according to the value of - log10(Q-value). See lines 239 to 243.
The protein-protein interaction results are not mentioned in the material and methods section.
Reply: We described this in2.4. Bioinformatic Analysis of DE mRNA, miRNA and lncRNA and Their Interactions, see lines 157 to 159.
The authors did not describe the comparison between the miRNAs and the target mRNAs presented on lines 339-340 in the material and methods section. The selected miRNAs and the mRNAs used for this analysis should be clearly shown and described in the material and methods section. The reason why these RNAs were selected should be clearly stated as well.
Reply:The interaction of miRNAs with mRNAs and lncRNAs were predicted with miRDB (http://www.mirdb.org/), miRcode (http://www.mircode.org/index.php) and TargetScan (http://http://www.targetscan.org/ vert_71/), which is described in Materials and Methods. Two mRNA-miRNA regulatory networks were constructed according to gene expression patterns in the LD muscle of LW pigs. According to this result, the expression of these DE mRNA genes were validated by RT-qPCR.
Discussion:
In general, the discussion is short and doesn't have enough focus on the functional candidate mRNAs, miRNAs, and lncRNAs. Additionally, a better link between the candidate molecules, the enriched biological processes, and the phenotypic differences between the breeds should be presented. How the results obtained help to better understand the differences of IMF between the breeds?
Reply: Thank you for your suggestions. We supplemented the discussion, See lines 401 to 419, 443 to 463.
On lines 361-363, the authors cited a previous study from the research group where the transcriptome of LD of LW was analyzed from 60d to 400d. The authors should compare the results obtained in the current study with the results obtained in the previous study. This approach should help to better identify functional candidate genes.
Reply: Thank you for your attention. Due to that the work is performed only in LW pigs, we did not perform this comparison. Besides, the tissues used for these two studies were different. We will test some of the DE mRNA genes in LW pigs aged from 60 d to 400 d.
From lines 381-389 the authors compared the number of DEGs from different studies. I suggest the authors remove this comparison and focus more on the candidate genes and networks obtained in the current study.
Reply: Thank you for your advice. We rewrote this as listed above.
The conclusion is poorly written. The authors should use the conclusion to describe how the obtained results fill up the current gap in the literature regarding the knowledge about the biological basis of IMF content in pigs. Additionally, perspectives and limitations could be cited.
Reply: We supplement the conclusion, see lines 508 to 518.
Minor comments:
Line 32: "invaluable results". This is speculative.
Reply: We removed " invaluable”. See line 33.
Lines 52-53: Insert space before reference square brackets.
Reply: This question has been modified. See line 55.
Lines 60-61: The authors mentioned "several studies" but only two were cited.
Reply: We replace "Several" with "Two". See line 63.
Line 132: Introduce space between by and RT-PCR.
Reply: This question has been modified. See line 129.
Line 162: Introduce space between RNAplex and the citation.
Reply: This question has been modified. See line 165 to 166.
Line 301-308. The sentence is very long.
Reply: We have modified the description in the article, see line 309 to 318.
Table 9 should be moved to Supplementary material.
Reply: Table 9 was moved to Supplementary material.
Line 377: Include reference about NAD+.
Reply: We added a reference about NAD+, see line 397.
Line 380: Introduce space between sows and the reference.
Reply: This question has been solved. See line 400.
Line 399: Avoid using "much higher" and show the quantitative value.
Reply: We used "higher" instead of "much higher". See line 429.
Line 405: Reference.
Reply: We did not find the reference and delete this sentence.
Lines 434-435: Which trans-target-genes?
Reply: Trans-target-genes enriched in AMPK pathway are IRS1 and SUGT1 and enriched in adipocytokine signaling pathway are IRS1 and LEPR, See line 486 to 492.

Reviewer 3 Report
The manuscript titled: “Comparative Transcriptomic Analysis of mRNAs, miRNAs and lncRNAs in the longissimus dorsi Muscles between Fat-type and Lean-type Pigs” analyzes the expression profile of mRNAs, miRNAs and lncRNAs in longissimus dorsi muscle of two pig breeds divergent for intramuscular fat content. They carried out differential expression analyses that reveal a total of 225 mRNAs, 12 miRNAs and 57 lncRNAs differentially expressed between Laiwu (LW, fat type) and Duroc x Landrace x Yorkshire (DLY, lean type) pigs. The study identified several genes that could be involved in intramuscular fat deposition. Although the study is interesting since they analyze the relationship among several types of RNA molecules and their implications with IMF deposition, the authors can get more out of their study and a thorough revision have to be carried out.
In a first place, the manuscript has several English mistakes, the order of appearance of the tables is not correct and some abbreviations have to be defined at the beginning of the manuscript.
Now, I am going to detail all the comments that have to be addressed before considering the manuscript for publication.
Line 14. “We performed RNA-seq analysis on the whole-transcriptomes”. Although the authors identify three different RNA molecules, they do not characterize the whole transcriptome since other RNA molecules such as tRNA, piRNA, rRNA, snoRNA … etc. are not identified in the present study.
Line 30. The definition of ceRNA appears for the first time in the manuscript at the end of it. If this term is not defined in the abstract it should be defined at least in the introduction.
Lines 37-38. “Pork is a vital source of protein, energy and iron for humans, accounting for about 43% of meat consumption in the world [1].” The citation that the authors use here is not correct. Ayuso et al use a similar sentence in their article to introduce their research, but they do not probe this fact, they use an adequate citation for making that affirmation. Please, use a convenient citation for this sentence.
Line 89. In my opinion, three biological replicates constitute a small sample size to carry out the study. More biological replicates should be included.
Line 132. “byRT-qPCR (TaqMan Probe).” Insert blank space between by and RT-qPCR.
Line 146. “and annotated with NCBI genome assembly [32]”. Please, indicate the exact version of the reference assembly and annotation files used to carry out the mapping and assembly. Do you also use HISAT2 for miRNA mapping? In that case, do you fit the same parameters? Why do not use a specific bioinformatics pipeline such as miRDeep2 to carry out the analysis?
Lines 150-162. Bioinformatic Analyses. Do you carry out three independent differential analyses expression, one per each RNA type? If so, please indicate in the description of the methodology.
Figure 3 shows two plots made with STRING and IPA software. This software are not mentioned or cited in the Bioinformatic Analyses section. I think they do not include also the tools used to make the plots that show KEGG enrichment. The authors have to include all the tools used in the study in the methods sections. They also have to include the parameters used to carry out the analyses. This is very important for the reproducibility of the research.
Line 175. There is an error in the order of the tables. Table 9 cannot be the first that appears in the manuscript, the first Table have to be Table 1. But I do not think the authors should re-numerate the tables, in my opinion “Table 9” is not such as relevant to be in the main text and it should be included as supplementary table.
Lines 224-225. “(Table 4) and LOC110258854 (downregulated) (Table 5), respectively.” There is an error about the tables here, table with the top 35 upregulated mRNAs is Table 3, and LDHB appears in Table 3 and the Table 4 is the one that shows the top 35 downregulated mRNAs and LOC110258854 appears in Table 4. Please, check and correct the citation and the order of the tables.
Lines 237-238, 277-278 and 309-311. In my opinion, the figures that show the expression of the RNA molecules quantified with RT-qPCR and with RNA-seq technique are not enough to demonstrate the validation of the mRNAs, miRNAs and lncRNAs, especially because they just use three biological replicates for the sequencing of the transcriptome. They should also show the correlation coefficients between both techniques and also to calculate a statistic such as the concordance correlation coefficient (Miron et al, 2006) to validate the entire experiment. This is necessary to actually check if RNAseq technique is validated.
Figures 2, 3, 4, 5, 6 and 7. The authors have to improve these figures. The plots are very small and they have a very low quality. It is is practically impossible to distinguish anything in the images. They should increase the size and maybe to pass some of the figures to supplementary material.
Line 291. “Totally 57 DE lncRNAs, 32 upregulated and 25 downregulated”. Please, replace totally by a total.
Discussion.
In my opinion, the authors do not properly discuss the results. For example, lines 360-361 are a new summary of the study that they carry out. In addition to this, the mention to the number of differential expression genes detected in other studies (lines 381-389) is not relevant. I think the authors should focus their discussion in the functional implications of the differentially expressed RNA molecules and the interaction among them to increase the insights about the biological processes underlying IMF deposition.
Miron, M., Woody, O. Z., Marcil, A., Murie, C., Sladek, R., and Nadon, R. (2006). A methodology for global validation of microarray experiments. BMC Bioinformatics 7:333. doi: 10.1186/1471-2105-7-333
Author Response
The manuscript titled: “Comparative Transcriptomic Analysis of mRNAs, miRNAs and lncRNAs in the longissimus dorsi Muscles between Fat-type and Lean-type Pigs” analyzes the expression profile of mRNAs, miRNAs and lncRNAs in longissimus dorsi muscle of two pig breeds divergent for intramuscular fat content. They carried out differential expression analyses that reveal a total of 225 mRNAs, 12 miRNAs and 57 lncRNAs differentially expressed between Laiwu (LW, fat type) and Duroc x Landrace x Yorkshire (DLY, lean type) pigs. The study identified several genes that could be involved in intramuscular fat deposition. Although the study is interesting since they analyze the relationship among several types of RNA molecules and their implications with IMF deposition, the authors can get more out of their study and a thorough revision have to be carried out.
In a first place, the manuscript has several English mistakes, the order of appearance of the tables is not correct and some abbreviations have to be defined at the beginning of the manuscript.
Now, I am going to detail all the comments that have to be addressed before considering the manuscript for publication.
Reply: Thank you for your comments. We corrected these errors.
Line 14. “We performed RNA-seq analysis on the whole-transcriptomes”. Although the authors identify three different RNA molecules, they do not characterize the whole transcriptome since other RNA molecules such as tRNA, piRNA, rRNA, snoRNA … etc. are not identified in the present study.
Reply: You are quite right. We checked the whole text and deleted “whole-“.
Line 30. The definition of ceRNA appears for the first time in the manuscript at the end of it. If this term is not defined in the abstract it should be defined at least in the introduction.
Reply: We put the definition of ceRNA in the introduction and cite related research. See lines 71 to 77.
Lines 37-38. “Pork is a vital source of protein, energy and iron for humans, accounting for about 43% of meat consumption in the world [1].” The citation that the authors use here is not correct. Ayuso et al use a similar sentence in their article to introduce their research, but they do not probe this fact, they use an adequate citation for making that affirmation. Please, use a convenient citation for this sentence.
Reply: You are right. We use a convenient citation to quote this sentence, see line 40.
Line 89. In my opinion, three biological replicates constitute a small sample size to carry out the study. More biological replicates should be included.
Reply: Thank you for your comment. You are right. Three samples for each group are the minimum requirement of transcriptome comparision. To reduce variations within group, the three samples with similar age, live weight and IMF were used.
Line 132. “byRT-qPCR (TaqMan Probe).” Insert blank space between by and RT-qPCR.
Reply: We inserted blank space between by and RT-qPCR. See line 129.
Line 146. “and annotated with NCBI genome assembly [32]”. Please, indicate the exact version of the reference assembly and annotation files used to carry out the mapping and assembly. Do you also use HISAT2 for miRNA mapping? In that case, do you fit the same parameters? Why do not use a specific bioinformatics pipeline such as miRDeep2 to carry out the analysis?
Reply: We used Bowtie2 software for miRNA mapping. This is parameter description: http://bowtie-bio.sourceforge.net/bowtie2/index.shtml
See lines 140 to 141.
Lines 150-162. Bioinformatic Analyses. Do you carry out three independent differential analyses expression, one per each RNA type? If so, please indicate in the description of the methodology.
Reply: We used DEseq2 software for differential expression analysis. The parameter settings are set according to the software instructions. See lines 145 to 147.
Figure 3 shows two plots made with STRING and IPA software. This software are not mentioned or cited in the Bioinformatic Analyses section. I think they do not include also the tools used to make the plots that show KEGG enrichment. The authors have to include all the tools used in the study in the methods sections. They also have to include the parameters used to carry out the analyses. This is very important for the reproducibility of the research.
Reply: We added the STRING, IPA software and KEGG enrichment methods to the “Bioinformatic Analyses of DE mRNA, miRNA and lncRNA and Their Interactions” sub-section in the material and methods. See lines 153 to 155, 157 to 160.
Line 175. There is an error in the order of the tables. Table 9 cannot be the first that appears in the manuscript, the first Table have to be Table 1. But I do not think the authors should re-numerate the tables, in my opinion “Table 9” is not such as relevant to be in the main text and it should be included as supplementary table.
Reply: Thank you for your suggestion. We deleted Table9 and submitted the content as supplement.
Lines 224-225. “(Table 4) and LOC110258854 (downregulated) (Table 5), respectively.” There is an error about the tables here, table with the top 35 upregulated mRNAs is Table 3, and LDHB appears in Table 3 and the Table 4 is the one that shows the top 35 downregulated mRNAs and LOC110258854 appears in Table 4. Please, check and correct the citation and the order of the tables.
Reply: Thank you for your attention. This errors are corrected. See lines 233 to 234.
Lines 237-238, 277-278 and 309-311. In my opinion, the figures that show the expression of the RNA molecules quantified with RT-qPCR and with RNA-seq technique are not enough to demonstrate the validation of the mRNAs, miRNAs and lncRNAs, especially because they just use three biological replicates for the sequencing of the transcriptome. They should also show the correlation coefficients between both techniques and also to calculate a statistic such as the concordance correlation coefficient (Miron et al, 2006) to validate the entire experiment. This is necessary to actually check if RNAseq technique is validated.
Reply: We corrected the statistics by using correlation methods. See Figures 2G, 4G and 6K.
Figures 2, 3, 4, 5, 6 and 7. The authors have to improve these figures. The plots are very small and they have a very low quality. It is is practically impossible to distinguish anything in the images. They should increase the size and maybe to pass some of the figures to supplementary material.
Reply: The original picture becomes blurred when it is put into the word file. We transfer the original picture into a separate file and name “Figure”.
Line 291. “Totally 57 DE lncRNAs, 32 upregulated and 25 downregulated”. Please, replace totally by a total.
Reply: Totally was replaced by a total. See line 299.
Discussion.
In my opinion, the authors do not properly discuss the results. For example, lines 360-361 are a new summary of the study that they carry out. In addition to this, the mention to the number of differential expression genes detected in other studies (lines 381-389) is not relevant. I think the authors should focus their discussion in the functional implications of the differentially expressed RNA molecules and the interaction among them to increase the insights about the biological processes underlying IMF deposition.
Reply: We added a discussion of genes associated with IMF deposition and muscle development. See lines 407 to 410, 447 to 451, 486 to 492.

Reviewer 4 Report
Please include diets used in this study.
Are LW pigs usually slaughtered with 50 kg?
The main concern regarding this study relates to the small sample size.
Could the fact that the animals were raised in different conditions (locations) affect the results of this research?
The criterion for slaughter was the age of 150 days. It is evident that the difference in the growth rate between breeds is large. Could this have had an impact on the results of the study?
Author Response
Please include diets used in this study.
Reply: We added this information. See lines 384-385.
Are LW pigs usually slaughtered with 50 kg?
Reply: LW pigs usually slaughtered with 80-90 kg. Here, we selected LW pigs according to its age, 150 days. Our previous study indicated a rapid fat deposition from 120 d to 240 d.
The main concern regarding this study relates to the small sample size.
Reply: Thank you for your comment. You are right. Three samples for each group is the minimum requirement of transcriptome comparision. To reduce variations within group, the three samples with similar age, live weight and IMF were used.
Could the fact that the animals were raised in different conditions (locations) affect the results of this research?
Reply: In the Discussion section, we addressed the efforts we did to reduce the effects of other factors except genetic components. See the second paragraph of Discussion section, lines 379 to 392.
The criterion for slaughter was the age of 150 days. It is evident that the difference in the growth rate between breeds is large. Could this have had an impact on the results of the study?
Reply: Thank you for your comment. You are quite right. The difference in the growth rate between breeds (LW and DYL) is large. This difference is caused by many factors, of which genetic background plays an important role. The purpose of this study is to find genes that may explain these differences in phenotype. In the Discussion section, we addressed the efforts we did to reduce the effects of other factors except genetic components. See the second paragraph of Discussion section, see lines 379 to 392.

Round 2
Reviewer 1 Report
The authors have made sufficient efforts to adequately address the reviewer's comments
Reviewer 2 Report
The authors provided a reviewed version of the manuscript entitled "Comparative Transcriptomic Analysis of mRNAs, miRNAs and 2 lncRNAs in the longissimus dorsi Muscles between Fat-type 3 and Lean-type Pigs". Some improvement in the information presented in the manuscript could be observed. However, there are still points to be substantially improved.
First, I would like to comment that the authors should provide complete answers to the reviewer’s comments. Most of the answers provided by the authors consist of one sentence and don’t discuss properly the concerns raised during the review process.
The authors modify the material and methods section trying to address the previous comments. However, little improvement was observed. For example, the authors were questioned about the multiple testing adjustment in the enrichment analysis. In the answer (and in the new version of the manuscript) the authors provided the link of an R package used to perform the analysis. This is not the proper answer. The authors must provide which kind of test was performed (FDR, Bonferroni, etc…). Additionally, the packages just be nominally cited, and the proper reference must be provided when available. For example, in the case of qvalue package, the authors mentioned that “Multiple testing correction of p value was applied using R package”. Where is the name of the package?
The authors were questioned about:
“Why the top 35 up and downregulated genes were presented? Is there any specific reason for this number?”
And the following answer was provided:
“Reply: Generally top 10 was listed. Some study listed top 50. Top 35 is not strictly and specifically selected. The reason for this selection is to show mor information.”
This is not a valid justification. The authors could provide a summary of general distribution of the DEG or define a threshold to select which genes to show. Otherwise, it is an arbitrary selection.
The authors still don’t provide the proper description of the reference used to align the reads during the RNA-seq analysis. Just providing a link in the manuscript is not enough. This link might be not available in the future. Please, provide the proper description with the name and version.
About the results from the Mann-Whitney U test provided by the authors. The significance level in Figure 1 was changed. Why? Was the data was reanalyzed? If yes, I would like to better understand the reason.
The authors were questioned about: “On lines 361-363, the authors cited a previous study from the research group where the transcriptome of LD of LW was analyzed from 60d to 400d. The authors should compare the results obtained in the current study with the results obtained in the previous study. This approach should help to better identify functional candidate genes.”
And the following answer was provided:
“Reply: Thank you for your attention. Due to that, the work is performed only in LW pigs, we did not perform this comparison. Besides, the tissues used for these two studies were different. We will test some of the DE mRNA genes in LW pigs aged from 60 d to 400 d.”
If the studies can’t be compared, there is no point to mention this study in the discussion.
The authors were questioned about: “On lines 230-231 the authors informed that the mRNAs upregulated in the LD muscle of LW pigs are enriched for several GO terms. However, genes/mRNAs/proteins/etc. are not enriched in this kind of analysis. The terms are enriched.”
And the following answer was provided:
“Reply: We only listed some terms according to the value of - log10(Q-value). See lines 239 to 243.”
This highlight that the authors didn’t understand properly the comment. Here, I was reinforcing a conceptual mistake from the authors. Genes are not enriched in a GO term/KEGG pathway analysis. The terms associated with these genes are enriched. The p-values are assigned for the terms, the statistical tests use the terms as a unit of comparison. I was not questioning the number of genes or thresholds. I was requiring a change in the text to remove a conceptual mistake.
In general, the new information provided must be checked carefully. There are several typos across the manuscript and there are some parts where sentences are terminated before the proper conclusion (by a misplaced period), for example on Lines 172-173.
The conclusion is still poor. The authors included information summarizing the number of DEG. Consequently, converting the conclusion into a list of results. The authors must provide an interpretative summary of the results obtained and how these results fill the gap in the current literature. The strong points and weaknesses of the study, as well as the further applications, should be presented.